# A solar optical hyperspectral library of rare earth-bearing minerals, rare earth oxide powders, copper-bearing minerals and Apliki mine surface samples

Friederike Koerting[1], Nicole Koellner[1], Agnieszka Kuras[1], Nina Kristin Boesche[1], Christian Rogass[1], Christian Mielke[1], Kirsten Elger[1], Uwe Altenberger[2]

[1] GFZ German Research Centre for Geosciences, Potsdam, 14473, Germany
[2] University of Potsdam, Institute of Geosciences, Potsdam, 14476, Germany

Correspondence to: Friederike Koerting (koerting@gfz-potsdam.de)

**Abstract.** Mineral resource exploration and mining is an essential part of today's high-tech industry. Elements such as rare earth elements (REE) and copper are, therefore, in high demand. Modern exploration techniques from multiple platforms (e.g. space- and airborne), to detect and map the spectral characteristics of the materials of interest, require spectral libraries as an essential reference. They include field and laboratory spectral information in combination with geochemical analyses for validation. Here, we present a collection of REE- and copper-related hyperspectral spectra with associated geochemical information. The libraries contain reflectance spectra from rare earth element oxides, REE-bearing minerals, copper-bearing minerals and mine surface samples from the Apliki copper-gold-pyrite-mine in the Republic of Cyprus. The samples were measured with the HySpex imaging spectrometers in the visible near infrared (VNIR) and short wave infrared (SWIR) range (400 – 2500 nm). The geochemical validation of each sample is provided with the reflectance spectra. The spectral libraries are openly available to assist future mineral mapping campaigns and laboratory spectroscopic analyses. The spectral libraries and corresponding geochemistry are published via GFZ Data Services with the following DOIs: http://doi.org/10.5880/GFZ.1.4.2019.004 (13 REE-bearing minerals and 16 oxide powders, Koerting et al. (2019a)), http://doi.org/10.5880/GFZ.1.4.2019.003 (20 Copper-bearing minerals, Koellner et al. (2019)), and http://doi.org/10.5880/GFZ.1.4.2019.005 (37 Copper-bearing surface material samples from the Apliki copper-gold-pyrite mine in Cyprus, Koerting et al. (2019b)). All spectral libraries are united and comparable by the internally consistent method of hyperspectral data acquisition in the laboratory.

# 1. Introduction

Reflectance spectroscopy is based on measuring the reflected solar radiation from a material of interest. It uses photosensitive detectors to record and analyse light reflected or scattered from the surface. The spectrum of the reflected light is unique for each material and acts like a spectral "fingerprint". Spectral libraries are comprehensive collections representing optical properties of materials in a specific wavelength range. In this data collection, hyperspectral spectra were collected under standardized laboratory or field conditions, include geochemical analyses of the sampled minerals and materials. The geochemical analyses can be used to check and interpret the hyperspectral spectra. Spectral libraries are essential in the field of imaging reflectance spectroscopy for mapping purposes. For example, the spatial distribution of ore-related mineral phases can be mapped by comparing unknown reflectance pixel spectra with known reflectance material spectra from a spectral library. The data that are being analysed are hyperspectral data cubes that are collected by e.g. satellite, UAV or tripod platforms to detect and map element or mineral occurrences in natural and in man-made surfaces.

The distinction of different surface materials or minerals is based on the nature of their reflectance spectral characteristics. The recorded reflectance spectral information is a function of the chemical and physical properties of the target material which cause different reactions to the incoming light on a molecular and atomic level (Clark, 1999; Hunt, 1982).

Spectral sensors collect the number of photons that are emitted or reflected per wavelengths by the material in each measured ground pixel. The interaction of the incoming light or radiant flux in a specific wavelength with the matter can reveal important information about the matter itself (Jensen, 2010). This interaction can be the absorption of a photon of a discrete energy state by an isolated atom or ion. This changes the atom's or ion's energy state. During this process energy is emitted that is not equal to the discrete energy of absorption which causes emissions at a different wavelength and creates "absorption bands" or "absorption features" (Clark, 1999; Hunt, 1982). The absorption feature position, depth and width depend on the different absorption processes taking place, the kind of chemical bond, the elements involved and the absorbing ion or molecule and its position in the crystal lattice, Absorption features in the visible and near infrared (VNIR: 400 to 1000nm) and short-wave infrared (SWIR: 1000 to 2500nm) wavelength region are caused by electronic and vibrational processes within the molecule or crystal lattice. The position and cause of these reflectance absorption features is discussed in detail e.g. in Clark (1999), Clark (2003) and Hunt (1982).

Hyperspectral data of geological surfaces can be acquired by ground- or UAV-based outcrop scans to map an ore body's surface mineral distribution by using spectral references libraries. An example of a hyperspectral surface mapping is shown in Figure 1. Here, the outcrop of former copper-gold-pyrite mine Apliki in the Republic of Cyprus was scanned hyperspectrally and mapped utilizing a spectral library of expected surface minerals. The analysis is based on United States Geological Survey (USGS) reflectance spectra. As the USGS spectral library entries do not origin from the same sensor as the mine face scan (HySpex data), they need to be spectrally adapted to the HySpex sensor properties.

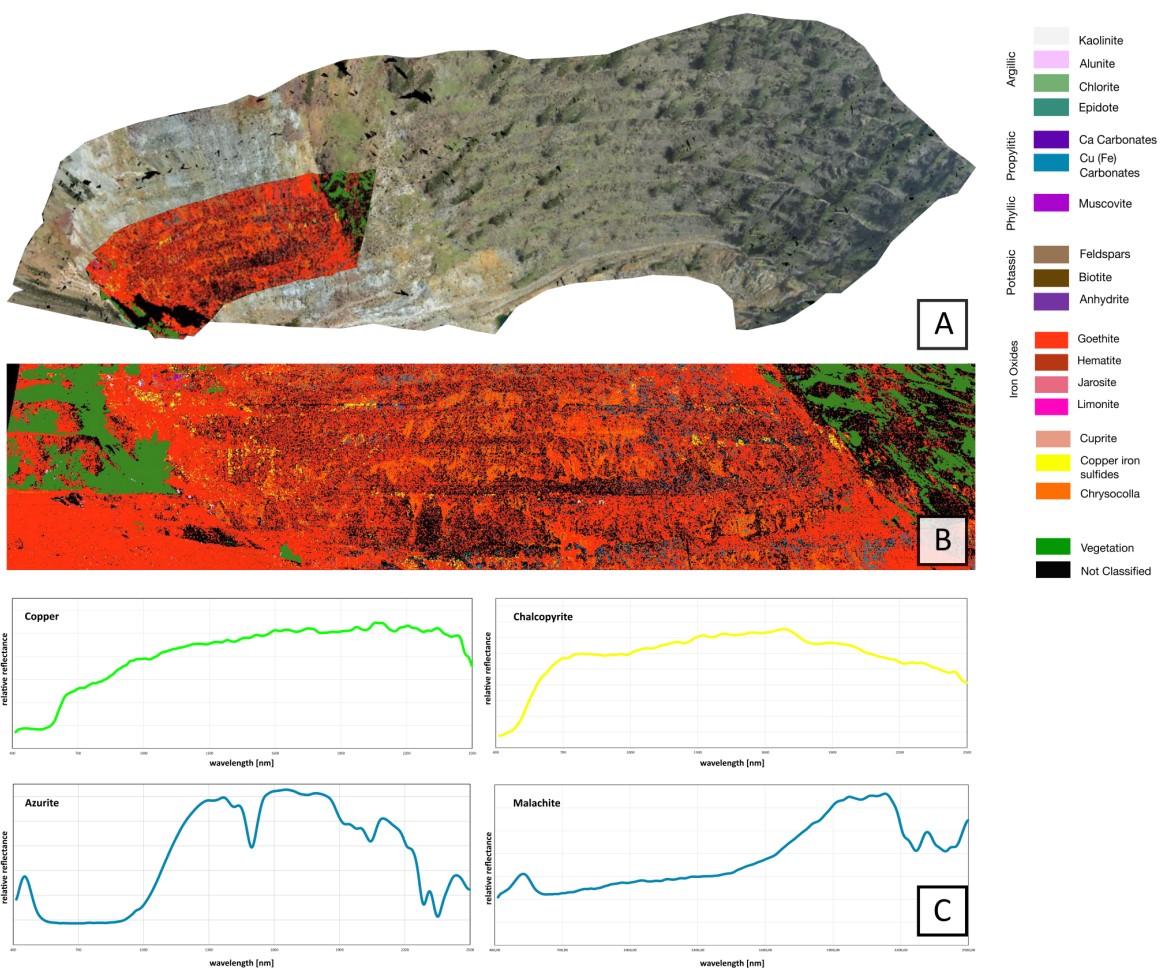

**Figure 1: Example for the application of a spectral library. A) 3D model of the open pit Apliki mine in the Republic of Cyprus based on RGB images and a superimposed analysis result of a hyperspectral HySpex scan. The hyperspectral map of the spatial mineral distribution from B) is stacked on the 3D model for visualization purposes. B) Analysis of a HySpex scan using a custom-made spectral library from USGS spectra (Clark et al., 2007). C) Example of hyperspectral spectra from copper-bearing minerals as presented in (Koellner et al., 2019).**

In the case of minerals reflectance spectra, only hyperspectral sensors with a spectral bandwidth resolution of approximately 10nm or less can capture the fine differences in reflectance at certain wavelength positions (Jensen, 2010). Future hyperspectral imaging satellites will provide the necessary data quality requirements to successfully map rare earth elements (REEs), copper deposits and other resources from space. These satellites will play an important role in the future of geological exploration, to help mapping large mineralized areas in remote regions (Mielke et al., 2016; Swayze et al., 2014). Several global mapping satellite missions will be launched in the next few years. Amongst them are the German EnMAP, the Chinese CCRSS-A and the Japanese HISUI missions (Guanter et al., 2015; Iwasaki et al., 2011; Tong et al., 2014). For those missions, the imaging spectroscopy community is currently developing methodologies e.g. for the detection of REEs in the image spectra (Boesche et al., 2015; Boesche, 2015; Bösche, 2015; Herrmann, 2019; van der Meer et al., 2012; Turner et al., 2014a, 2014b; Turner, 2015).

We aim to contribute to the already existing, accredited libraries, e.g. the USGS- and the ECOSTRESS Spectral Library and various others (Baldridge et al., 2009; Clark et al., 2007; Hunt, 1977; Kokaly et al., 2017; Meerdink et al., 2019; Percival et al., 2016). The available reflectance spectral libraries are commonly based on powdered natural or synthetic samples that are spectrally pure. The spectral data is usually collected by point-spectroradiometers e.g. the Analytical Spectral Device (ASD) FieldSpec® 3. Our contributed reflectance spectra are based on imaging spectroscopy data from the HySpex classic series scanning samples in a natural and a powdered state. Reflectance spectral libraries like the here presented, based on HySpex imaging data and untreated samples, are not yet freely available for the hyperspectral community.

The spectral and geochemical information of samples presented here, belong to three different mineral assemblages and correspond to three different types of deposits. The sample's spectral information is provided within four spectral library files and their corresponding geochemical composition files. The four spectral library files represent (1) REE-bearing minerals, (2) synthetic REE-oxide powders (Koerting et al., 2019a), (3) copper-bearing minerals (Koellner et al., 2019) and (4) powders of copper-bearing surface material from the Apliki copper-gold-pyrite mine in the Republic of Cyprus (Koerting et al., 2019b). Spectrally, the libraries cover the full wavelength range of the solar optical range (414 nm – 2498 nm). The corresponding geochemical analyses are explained in the methods for each sample type. The two REE libraries (Koerting et al., 2019a) consist of the spectra of 16 rare earth oxide (REO) powders and 13 REE-bearing minerals (REMin). In addition, the spectra of niobium- and tantalum oxide powders are provided, which will further not be mentioned individually but be included in the term "REO". The third spectral library includes 20 copper-bearing minerals (Koellner et al., 2019) and the fourth spectral library contains 37 surface samples from the Apliki copper-gold-pyrite mine site in the Republic of Cyprus (Koerting et al., 2019b). All spectral libraries are united and comparable by the internally consistent method of hyperspectral data acquisition in the laboratory. An extensive list of the samples can be found in the technical reports provided with each dataset.

The samples are presented as reflectance spectral libraries and their geochemical composition. Sample nominations are based on the geological collection of origin or sample abbreviations from the field sampling. The sample nomination is not an

110 interpretation of the presented geochemical data. The datasets are independent of each other, the reflectance spectra can be seen as a spectral expression of the existing geochemical data. Neither the geochemistry nor the reflectance spectra are interpreted or correlated to each other.

The outline of this document follows the necessary line of knowledge to successfully make use of the here presented spectral

libraries. Section 2 includes a description of the analysed materials and Section 3 informs about the methods including the sample preparation and spectra collection, the hyperspectral data acquisition, covering the processing of the data and spectral measurement parameters and the geochemical analyses of the samples. Section 4 lists the samples that were measured spectrally and geochemically and the data of which can be accessed via the GFZ Data Services platform. Section 5 discusses the parameters influencing the data. A separate data description and the geochemical analysis results are included as data

reports in the three different data publications (Koellner et al., 2019; Koerting et al., 2019a, 2019b).

## 2. Materials

The REE sample material includes 16 REO powders (REO) and 13 REE-bearing minerals (REMin). The REO powders belong to a series of rare earth metals and compounds (REacton®) and were purchased from Alfa Aesar. All REO powders contained

at least 99.9% of the REE oxide, as per the seller-supplied concentration certificates. The concentration certificate information can be found in the data description of (Koerting et al., 2019a). The REO powders were obtained as high-purity materials with a grainsize of <63 µm. The REMin samples (ore minerals) were purchased from Gunnar Färber Minerals, an online trader of mineral_specimens. The mineral notation is based on the sample name provided by Gunnar Färber Minerals. The supplier offers analytical services with a modern REM-EDX technology and therefore we assume the specimen are analysed and the

mineral species is validated before the sale. The X-Ray Fluorescence (XRF) data presented in the data description of (Koerting et al., 2019a) should be consulted to validate the given mineral nomination noted by Gunnar Färber Minerals.

The 20 copper-bearing minerals belong to collections of the University of Potsdam (UP) and the Federal Institute for Geosciences and Natural Resources (BGR), a samples list can be found in (Koellner et al., 2019). The minerals were measured hyperspectrally with no sample preparation, the sample photos and geochemical analysis are provided in the data description

for (Koellner et al., 2019). The 37 Apliki mine surface samples were collected (Koerting et al., 2019b) in March 2018 during a field campaign of the Geological Survey Department of the Republic of Cyprus (GSD) and the GFZ German Research Centre for Geosciences (GFZ). Surface material in the mine was collected and prepared (crushed and pulverized) for the geochemical analysis by Bureau Veritas Minerals (BVM). The powdered samples were measured hyperspectrally as powder tablets, a sample list including photos from the in-situ conditions of the samples can be found in the technical report (Koerting et al.,

2019b).

## 3. Methods

### 3.1 Sample Preparation and spectra collection

The sample preparation varies by sample type and depends on the material and the information of interest. This is based on the research projects that the samples stem from and for which the spectral and geochemical data was acquired.

The reflectance spectra for each sample were manually extracted from the processed hyperspectral image scenes by averaging a number of pixels over a central sample area. The resulting spectra were compiled in a spectral library. Thereby, each reflectance spectrum of a spectral library represents an average reflectance spectrum of the material, depending on the sample size and spectral homogeneity. The extraction of the reflectance spectra is explained in detail in each data description (Koellner et al., 2019; Koerting et al., 2019a, 2019b).

The REO powders were measured in 100% quartz glass petri dishes underlain by black cellular rubber, each powder was measured separately. Figure 2 shows the measurement setup of holmium-oxide powder as an example for the REO powders. The REE-bearing minerals were measured separately. Figure 3 shows the xenotime sample (brownish single crystal embedded in quartz) as an example for the REMin samples. The REMin samples were measured without sample preparation on black cellular rubber, as is shown for the copper-bearing minerals in Figure 4. For all measurements, the final reflectance spectral analyses were spatially reduced to the centre pixels of each identified REE-bearing mineral or a 5x5 pixel average reflectance spectrum centred on the REO powder sample. Shadow effects from the sidewalls of the boxes could thus be minimized. One representative reflectance spectrum of every REMin and REO sample was collected for the spectral library (Herrmann, 2019).

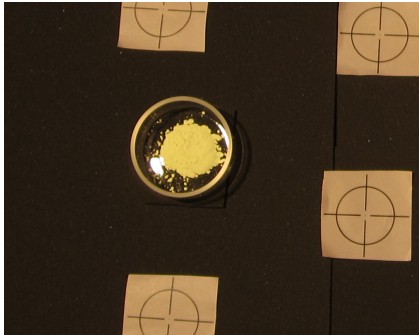

**Figure 2: Holmium-oxide powder in the laboratory HySpex setting in a quartz glass petri dish underlain by black cellular rubber. Geometric markers for the pre-processing were placed alongside the sample.**

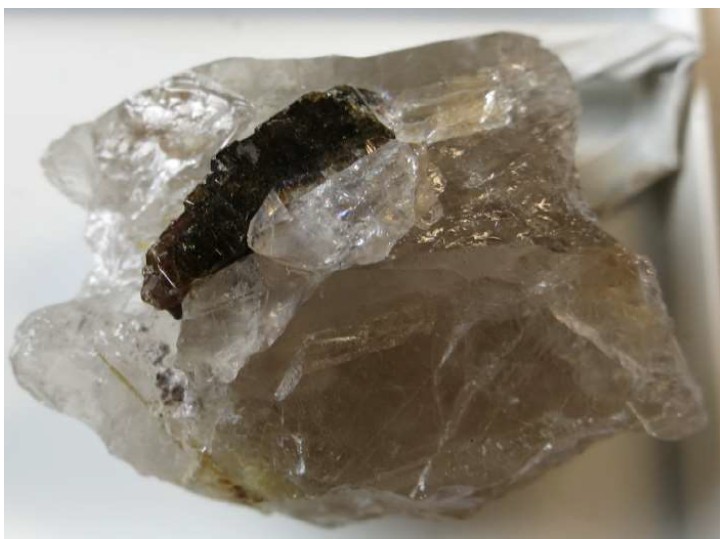

**Figure 3: Xenotime embedded in quartz as an example for the REE-bearing mineral samples.**

The copper-bearing mineral samples were measured without any sample preparation as the variable surface of the minerals and the influence of the mineral structure was of interest. Figure 4 shows an example scan of some of the copper-bearing minerals. The full sample list including sample photos and the marked area of the geochemical sampling can be found in the
technical report (Koellner et al., 2019). The area used to obtain the spectrum, averaging over a 5x5 pixel window, was afterwards sampled for the geochemical analysis.

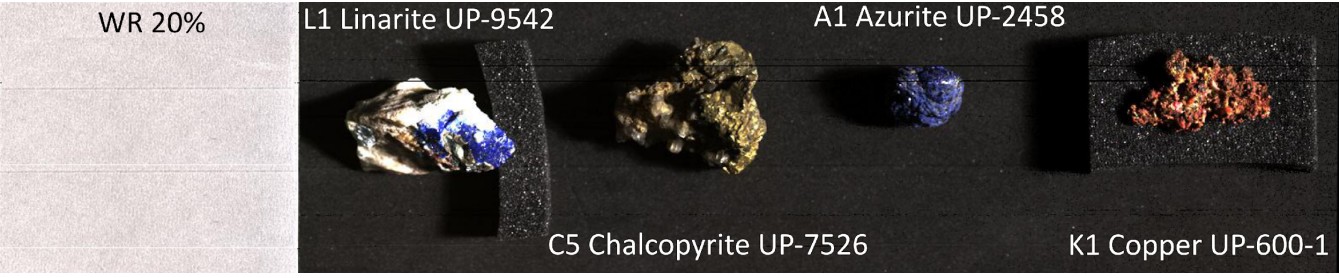

**Figure 4: Showing HySpex scan "MH_FK_LAB_Cudetect_008_09012018_WR20" as an example to highlight the lack of sample**
**preparation.**

The Apliki mine samples were crushed and powdered so that ≥85% of the sample was below 75µm. Homogenized powders were measured as pressed powder tablets (Figure 5). The area to obtain the sample's reflectance spectrum was chosen over a 5x5 pixel window in the centre of the powder tablet to minimize influences from the tablet's metal frame. The dark spots in
each tablet were caused by previous measurements with a laser induced breakdown spectrometer (LIBS). The hyperspectral

sample spots were chosen in order to exclude the measurement points of the LIBS in the spectral footprint. In case of broken powder tablets like "7d_Hem", the shadowed, rough surface areas were excluded from the spectral sampling and an even powder surface was favoured.

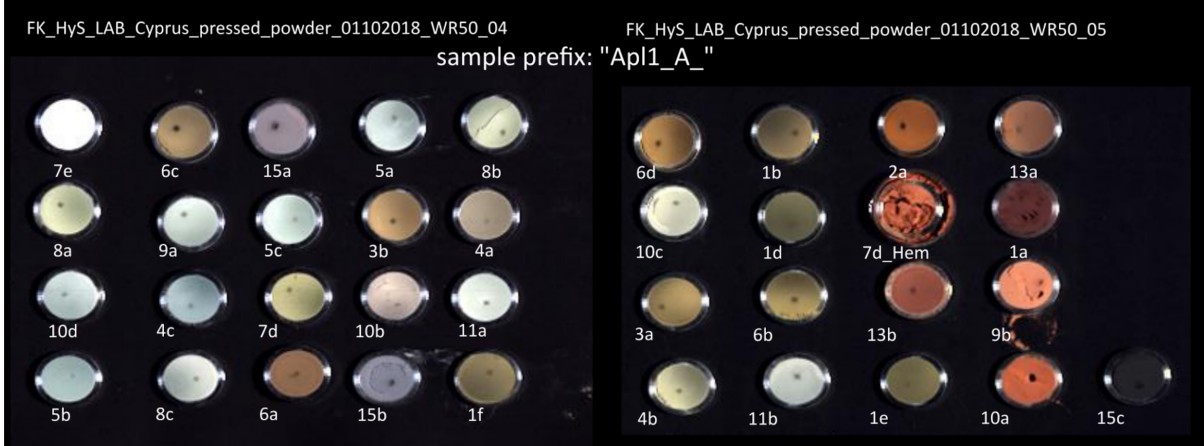


**Figure 5: Showing the Apliki mine samples prepared as powder tablets.**

### 3.2 HySpex Data Recording

The HySpex VNIR-1600 and SWIR-320m-e (technical description available at: (hyspex.no/products/disc.php, 2019)) are two
line-scanning cameras mounted in parallel. They cover the range of the visible to near infrared (VNIR, 414 – 993 nm) and the short-wave infrared (SWIR, 967 – 2498 nm) wavelength region. The sensors record an array-line of 1600 pixel (VNIR) and 320 pixel (SWIR) (push-broom scanning). Every pixel contains a spectrum with a total spectral sampling number of 408 bands in total.

The HySpex cameras are provided with two acquisition modes, one for airborne data collection and one for laboratory
measurements. In laboratory mode, the cameras are combined with a trigger pulse-moving sleigh (translation stage) of definable frame period (depending on the integration time of every array-line acquisition). The configuration of the translation stage framework, the cameras and the light source (Halogen GX6.35, 2 x 1000 W, 45° illumination angle) are fixed, while the sleigh and the samples are moving through the focal plane (Rogass et al., 2017).

The reflectance level of a white reference panel, placed in line with the samples, is chosen according to the albedo of the
samples. The higher the albedo of the sample, the higher is the diffuse reflectance factor of the white reference panel that is chosen. For the REE samples (REMin and REO), a white reference panel of 95% reflectance was used, because most of the REO samples were bright, white powders of a high albedo, this is based on test measurements of (Bösche, 2015; Herrmann, 2019). The Apliki samples required a 50% reflectance white reference panel, whereas the copper-bearing minerals were measured using a 20% reflectance white reference panel. Both, the geometrical setup and the heat up time of the lamp influence
the configuration of the light source. The maximum illumination was obtained with an angle of 45° between the incident light

and the vertical plane. The distance between the lamp and the HySpex cameras was higher compared to the distance between the samples and the sensor to ensure diffuse illumination and to avoid thermal influence on the cameras and the samples. The integration time (= measurement time for each image line) was tested to be as high as possible to suppress the impact of signal uncorrelated gaussian white noise and at the same time as low as needed to avoid detector saturation. For all measurements

the integration time was chosen with respect to the sample albedo. The HySpex sensor characteristics are listed in Table 1. The used settings for the REMin and REOs are listed in Table 2, the settings for the copper-bearing minerals in Table 3 and for the Apliki mine samples in Table 4. The laboratory is equipped with black-painted walls and doors, as well as black curtains to avoid reflected light from surfaces other than the sample, an example setup of the sensors, the translation stage and the samples can be seen in Figure 6. The laboratory conditions were kept stable, the air temperature was regulated to 21±0.5°C

and the humidity was below 70% for all measurements. Black cellular rubber is used as a base material for all samples for hyperspectral data acquisition. It reflects less than 5% on average of the incoming radiation.

Detailed descriptions for the GFZ' standard measurements and the process chain can be found in (Rogass et al., 2017).

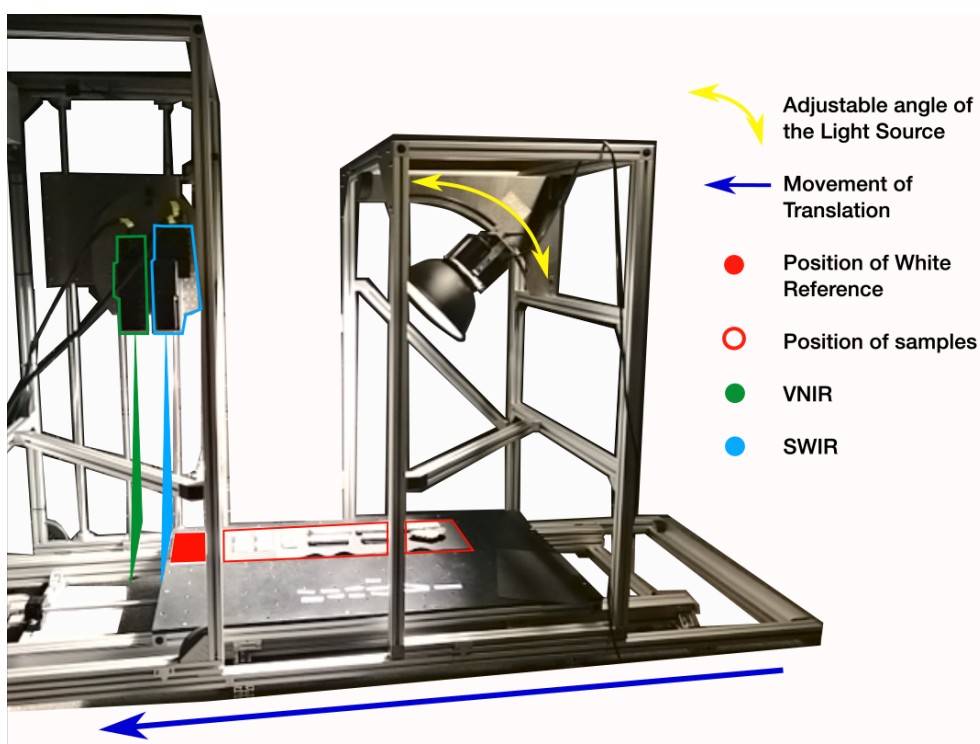


**Figure 6: The HySpex translation stage setup (Körting, 2019).**

**Table 1: HySpex sensor parameters of the VNIR-1600 (VNIR) and SWIR-320m-e (SWIR)**

| **HySpex** sensor parameters | | |
|---|---|---|
| Lamp arrangement | 45° | |
| | VNIR | SWIR |
| Wavelength range [nm] | 414 - 993 | 967 - 2498 |
| Pixels per line | 1600 | 320 |
| Sampling interval [nm] | 3.7 | 6 |
| Radiometric resolution | 12 bit | 14 bit |
| Light source | Halogen GX6.35, 2 x 1000 W | |


**Table 2: HySpex settings for laboratory measurements of the REO and REMin (Koerting et al., 2019a, modified after (Bösche, 2015; Herrmann, 2019).**

| **HySpex** settings | | |
|---|---|---|
| Distance, sample to sensor | 1 m | |
| Sensor arrangement head to head | 1m lenses, eq on VNIR | |
| | VNIR (1600 px) | SWIR (320 px) |
| Integration time [μs] | 30 000 | 5 000 |
| Frame period [μs] | 31 000 | 123 506 |

**Table 3: HySpex settings for laboratory measurements of the copper-bearing minerals (Koellner et al., 2019).**

| **HySpex** settings | | |
|---|---|---|
| Distance, sample to sensor | 30cm | |
| Sensor arrangement head to head | 30cm lenses, eq on VNIR | |
| | VNIR (1600 px) | SWIR (320 px) |
| Integration time [μs] | 120000 - 140000 | 15000 – 20000 |
| Frame period [μs] | 120062 - 141004 | 478334 - 561768 |


**Table 4: HySpex settings for laboratory measurements of Apliki mine powdered samples (Koerting et al., 2019b).**

| **HySpex** settings | | |
|---|---|---|
| Distance, sample to sensor | 1 m | |
| Sensor arrangement head to head | 1m lenses, eq on VNIR | |
| | VNIR (1600 px) | SWIR (320 px) |
| Integration time [μs] | 60000 | 10000 |
| Frame period [μs] | 60060 | 239282 |

### 3.3 Hyperspectral Data Processing

Each measurement run produces one VNIR and one SWIR 3D-data cube. The three dimensions are the two spatial x,y- and the spectral z-dimension. The 3D image cubes are produced, by moving a homogeneous reflecting white reference panel and the samples through the focal plane of the two sensors. The VNIR image cube is resized to the spatial dimensions of the SWIR data cube, co-registered and stacked with the SWIR data cube resulting in a continuous image cube with the spectral range of 414 – 2498 nm. In order to produce a reflectance image, the image pixel that show the white standard were averaged to a one-

line reference spectrum. The reflectance was calculated by dividing every image line spectrum by its reference spectrum from the reflecting white reference panel. The resulting reflectance data are scaled from 0 - 10000. A detailed description for the laboratory set-up and processing can be found in (Rogass et al., 2017). The software 'HySpex ground' was used to perform the measurements and the software 'HySpex rad' was used to perform the radiometric calibration on the image data.

### 3.4 Geochemical Sample Analysis for Sample Characterization

Depending on the sample type, the geochemical analysis methods differ. The methods used for each sample type, are listed in Table 5.

**Table 5: Sample type and corresponding geochemical characterization method.**

| Sample type | Geochemical Analysis |
|---|---|
| REO (Koerting et al., 2019a) | Laboratory certificates |
| REMin (Koerting et al., 2019a) | X-Ray Fluorescence (XRF), Electron probe microanalyzer (EPMA) analyses |
| Copper-bearing minerals (Koellner et al., 2019) | Scanning electron microscope (SEM), EPMA |
| Apliki mine samples (Koerting et al., 2019b) | Bureau Veritas Mineral Analysis, ICP-MS and ES |


### 3.4.1 Thermo Niton XL3t (XRF)

**REMin**

The geochemical measurements for the REMins were performed using an X-Ray Fluorescence (XRF) instrument - Thermo Niton XL3t (Fisher Scientific, 2002). The XL3t is a lightweight, hand-held XRF analyzer. The measurement principle follows

the principle of X-Ray fluorescence, where the sample inbound X-Rays excite electrons to a higher energy level in the sample material. Energy in form of XRF radiation is released when these electrons return to their original state. The frequency of this radiation is characteristic for the measured chemical element and its intensity is correlated to the concentration level. The intensity of each element is detected as counts per second by the detector, a geometrically optimized large area drift detector (GOLDD). The maximum excitation voltage of the XL3t device is 50 kV, which means out of the full REE suite only four

light REEs can be detected (Lanthanum, Cerium, Praseodymium and Neodymium).

The XL3t spectrometer is attached to a lead shielded sample chamber, in which samples with a diameter smaller than 3.3 cm can be placed. Mineral samples can be directly placed in the chamber; powdered samples have to be placed in sample tubes (2.5 cm diameter). The sample tubes are made of plastic with a plastic foil on the bottom. The plastic cannot be detected by XRF and therefore not interfere with the measurements. A built-in camera of the XL3t enables the precise location of the measuring spot. The software used for the measurements is named "NDTr" and the measurement mode was "mining and exploration". The concentration levels are provided along with a balance value. "Balance" represents counts per seconds that could not be attributed to one of the measured elements. Table 6 shows the measurement modes and filters used. In-depth description of the XL3t and the XL3t-results for each sample can be found in (Bösche, 2015; Herrmann, 2019)

**Table 6: Settings used for the Thermo Niton XL3t X-ray fluorescence device (Bösche, 2015).**

| **Thermo Niton XL3t** Setting | |
|---|---|
| Measurement mode | Test all geo |
| Filter | Main, Low, High, Light |
| Filter measurement time | 30 seconds each |

### 3.4.2 Scanning Electron Microscope (SEM) and Electron Probe Microanalyzer (EPMA)

**Copper-bearing minerals**

In order to obtain information about the zonation and internal fabrics of the copper-bearing minerals a fully automated JEOL JSM-6510 scanning electron microscope (SEM) (20kV acceleration voltage) at the University of Potsdam was used. A back-scattered electron detector displays compositional variation in the imaging area based on the mean atomic number of the pixel. An energy dispersive X-ray spectrometer (EDX, Oxford Instruments INCAx-act) attached to the instrumentation provides quantitative elemental analysis of single spots. After calibrating with pure copper, a wide spectrum of elements can

be identified. Based on previous results, divergences of up to 5 weight % can be expected, which for quantitative analysis is acceptable.

In order to approximate the values for copper a JEOL JXA-8200 electron probe microanalyzer (EPMA) at the University of Potsdam was used. The electron microprobe is equipped with five wavelength-dispersive X-ray spectrometers (WDX) and

was operated with a 20 kV accelerating voltage, a 20 nA current, and a beam diameter of 2 µm. The analytical counting times were 20/10 s for the element peak and 10/5 s for background positions. Analyses were calibrated using silicates/sulphides obtained from the Smithsonian Institution and Astimex. Quantifying elements of a lower atomic mass than boron is not possible, carbon cannot be measured either.

An example SEM analysis for copper-bearing mineral sample "C1_Chalcopyrite" can be seen in Figure 7, the EPMA analysis of the mineral is listed in Table 7. The full SEM and EPMA results are documented in (Koellner et al., 2019).

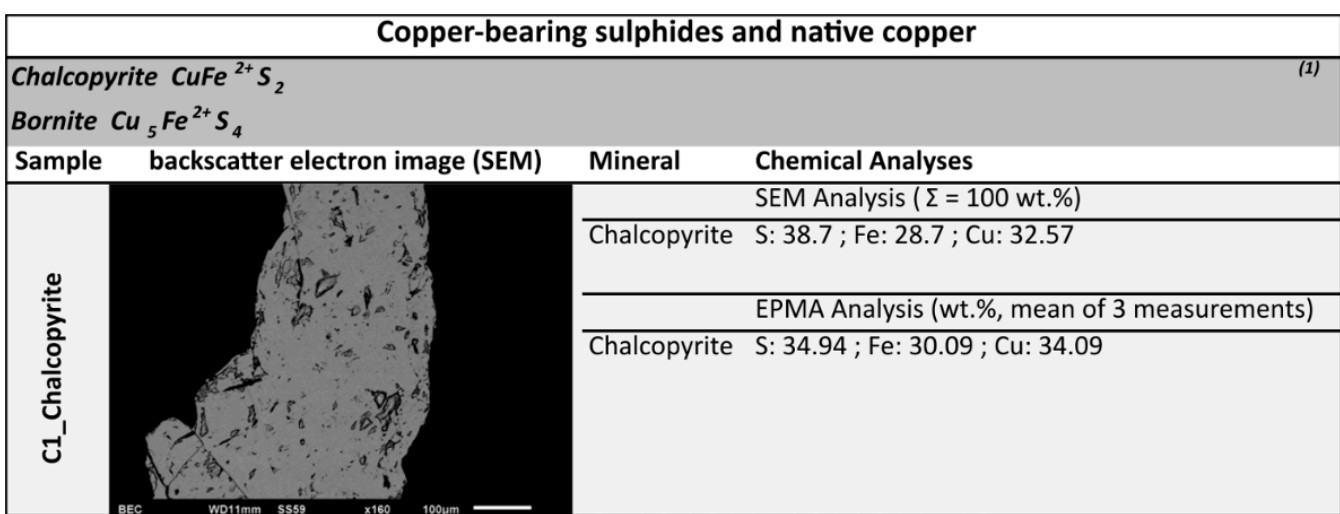

Figure 7: Sample C1_Chalcopyrite SEM and EPMA analysis.


**Table 7: Sample C1_Chalcopyrite EPMA analysis results, from three sample-points on the sample. Element concentrations reported in wt% or as "below detection limit" (bdl).**

| Sample-point | Al [wt%] | Hg [wt%] | Fe [wt%] | Cu [wt%] | Si [wt%] | S [wt%] | Mn [wt%] | Total [wt%] |
|---|---|---|---|---|---|---|---|---|
| C1_Chalcopyrite-1 | bdl | bdl | 30.00 | 33.98 | bdl | 34.81 | bdl | 98.79 |
| C1_Chalcopyrite-2 | bdl | bdl | 30.19 | 34.108 | bdl | 34.94 | bdl | 99.23 |
| C1_Chalcopyrite-3 | bdl | bdl | 30.08 | 34.194 | bdl | 35.09 | bdl | 99.36 |

## REE-bearing Minerals


Some of the REMin (xenotime, bastnaesite, fluorapatite, synchysite and ilmenite) were additionally analysed by using a JEOL JXA-8200 electron microprobe (EPMA) at the University of Potsdam based on a method developed by (Lorenz et al., 2019). The conditions used for the analysis were: 20kV acceleration voltage, 20nA beam current and a beam size of 2 μm. Counting times were between 10 s - 20 s on peak for major elements and 50 s for REE and other trace elements.

The following spectral lines and mineral standards from Smithsonian and Astimex were used: fluorapatite (F Kα, P Kα, Ca Kα), albite (Na Kα), fayalite (Fe Kα, Mn Kα), wollastonite (Si Kα), omphacite (Al Kα), LaPO₄ (La Lα), PrPO₄ (Pr Lβ), CePO₄ (Ce Lα), NdPO₄ (Nd Lβ), YPO₄ (Y Lα), EuPO₄ (Eu Lα), SmPO₄ (Sm Lβ), LuPO₄ (Lu Lα), GdPO₄ (Gd Lα), ErPO₄ (Er Lβ), DyPO₄ (Dy Lβ), YbPO₄ (Yb Lα), HoPO₄ (Ho Lβ), uranothorite (U Mβ), crocoite (Pb Mβ). The EPMA data were reduced

using the software-implemented PRZ-XXP data-correction routine, which is based on the φ(ρz) method (Heinrich and Newbury, 1991).

### 3.4.3 Apliki mine surface sample analysis

The Apliki mine samples were analysed by Bureau Veritas Minerals – Mineral Laboratories Canada (BVM) using their standard packages (Bureau Veritas, 2020). The samples were pulverized below 75 μm and analysed for major, minor and trace elements using ICP-MS and ES. The results are grouped by the interal BVMs sample preparation-/ analysis method types . Those "analysis method types" were namely "aquatic", "rock" and "soil". The sample numbers, associated analysis method, type and internal BVM analysis codes can be found in the technical report of the Apliki mine surface sample data (Koerting et al., 2019b).

## 4. Results

The following samples are provided and described in detail in the corresponding technical reports. For clarity purposes, all provided samples and corresponding spectra names are listed here, including a short sample description and, where applicable, the sampling location, geochemistry or mineralogy (Table 8-12). The detailed sample descriptions can be found in the corresponding technical reports provided with the data. For each file collection a plot of the spectral library is shown (Figure 8-13).

### 4.1 REE-bearing minerals and rare earth oxide powders

**Table 8: Samples, sample names and locality and spectral library filenames of REE-bearing minerals**

| Sample | Original sample name | Sample locality | Spectrum name |
|---|---|---|---|
| Aeg | Aegirine, "Acmite" | Rundemyr. Øvre Eiker. Buskerud. Norway/TYP | REMin_Aeg |
| Bar | Bariopyro-chlore. Fluorapatite | Mina Boa Vista. Catalao. Goias/Brazil | REMin_Bar |
| Bst | Bastnaesite (Ce) | Zagi Mountain. Warzal Dam. Pechawar. North-West Frontier Prov./Pakistan | REMin_Bst |
| Fap | Fluorapatite, Albite | Golconda Mine. Governador Valadares.  Doce Valley. Minas Gerais/Brazil | REMin_Fap |
| Flt | Fluorite | Arbegona. Shashemanne. | REMin_Flt |
| Gdl | Gadolinite (Y) Synchysite (Y), Fluorite | White Cloud Pegmatite. South Platte. Jefferson Co. Colorado/USA | REMin_Gdl |
| Ilm | Ilmenite | Mogok. Sagaing District. Mandalay/Myanmar | REMin_Ilm |
| Pcr | Polycrase (Y) | Puoutevare Pegmatite. Tjalmejaure Lake. Jokkmokk Lappland/Northern Sweden | REMin_Pcr |
| Prs | Parisite (Nd) incl. Parisite (Ce) | Mountain Pass Mine. Ivanpah Mts.  San Bernardino Co. California/USA | REMin_Prs |
| Syn | Synchysite (Y), Microcline, Quartz | White Cloud Pegmatite. South Platte. Jefferson Co. Colorado/USA | REMin_Syn |
| Xtm1 | Xenotime (Y) (a) | Novo Horizonte. Ibitiara. Bahia/Brazil | REMin_Xtm1 |
| Xtm2 | Xenotime (Y) (b) | Novo Horizonte. Ibitiara. Bahia/Brazil | REMin_Xtm2 |
| Zrn | Zircon | Peixe alkaline complex. Monteirópolis. Jaú do Tocantins. Tocantis/Brazil | REMin_Zrn |

**Table 9: Sample name and supplier, product and lot number and spectral library filenames of the rare earth oxide powders.**

| Sample name, Supplier | Product Number | Lot Number | Spectrum Name |
|---|---|---|---|
| Yttrium (III) oxide, Sigma-Aldrich | 204927 | MKBL2030V | REO_Yttrium |
| Niobium (V) oxide, Alfa Aesar | 11366 | L18Y022 | REO_Niobium |
| Lanthanum (III) oxide, Alfa Aesar | 11272 | B08X015 | REO_Lanthanum |
| Cerium (IV) oxide, Alfa Aesar | 11372 | L07S057 | REO_Cerium |
| Neodymium (III) oxide, Alfa Aesar | 11250 | C02W029 | REO_Neodymium |
| Samarium (III) oxide, Alfa Aesar | 11229 | 61200836 | REO_Samarium |
| Europium (III) oxide, Alfa Aesar | 11299 | A16Z001 | REO_Europium |
| Gadolinium (III) oxide, Alfa Aesar | 11290 | A13W016 | REO_Gadolinium |
| Terbium (III.IV) oxide, Alfa Aesar | 11208 | J24Q019 | REO_Terbium |
| Dysprosium (III) oxide, Alfa Aesar | 11319 | 61300733 | REO_Dysprosium |
| Holmium (III) oxide, Alfa Aesar | 11280 | J11X030 | REO_Holmium |
| Erbium (III) oxide, Alfa Aesar | 11310 | 61000356 | REO_Erbium |
| Thulium (III) oxide, Alfa Aesar | 11198 | F25S060 | REO_Thulium |
| Ytterbium (III) oxide, Alfa Aesar | 11191 | 61201069 | REO_Ytterbium |
| Lutetium (III) oxide, Alfa Aesar | 11255 | G14X082 | REO_Lutetium |
| Tantalum (V) oxide, Alfa Aesar | 14709 | I14Y039 | REO_Tantalum |

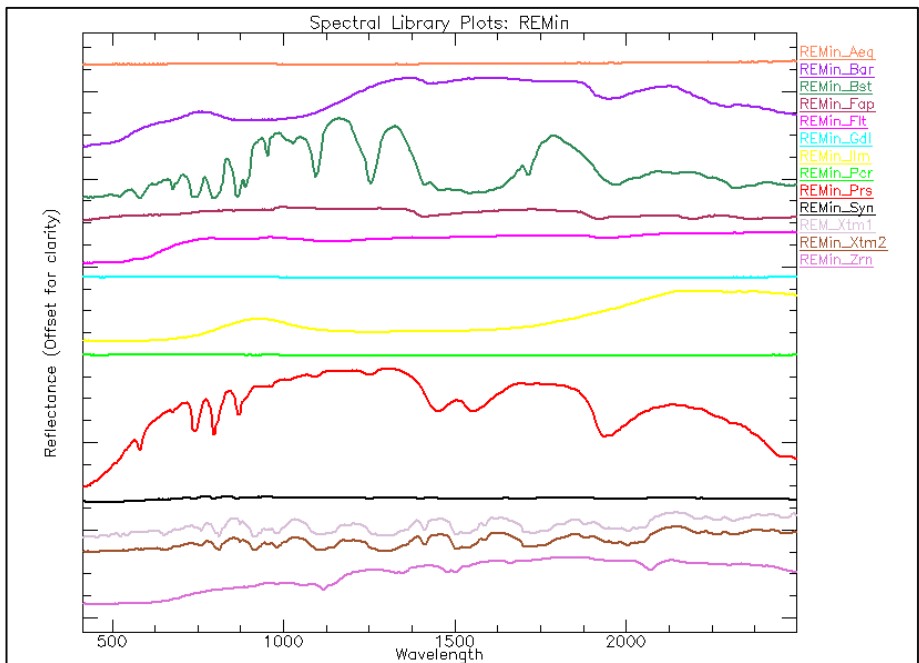

**Figure 8: Spectral library plot of the REE -bearing minerals.**


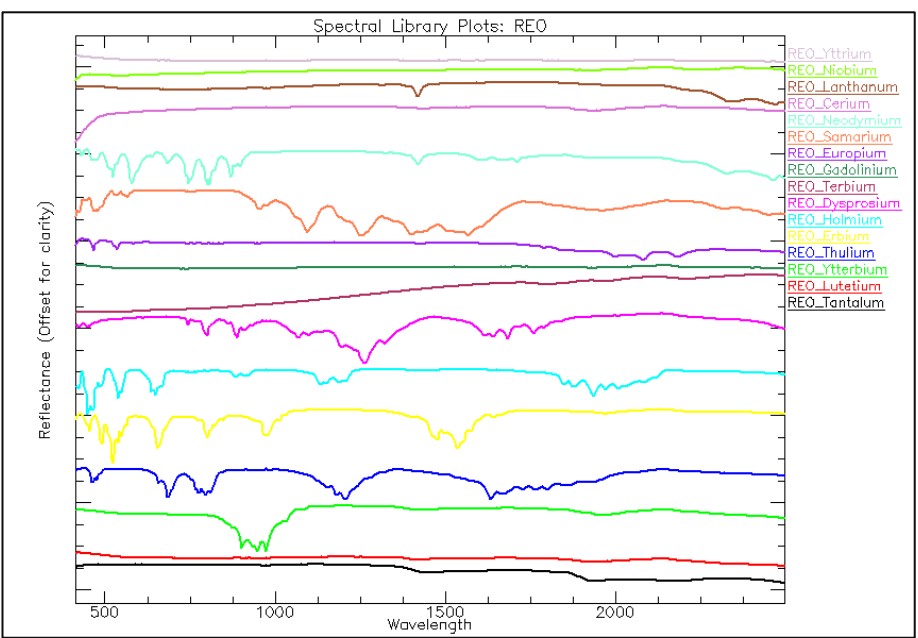

**Figure 9: Spectral library plot of the rare earth oxide powders.**

## 4.2 Copper-bearing minerals

**Table 10: Sample names, collection, original sample name, locality, alteration, mineral formula, spectral library filenames and geochemical composition of the copper-bearing sulphides and native copper**

| Sample name | Collection | Original sample name | Sample locality | Visible alteration | Spectra names | Geochemical composition (EPMA mean, n=3, wt%) |
|---|---|---|---|---|---|---|
| C1_Chalcopyrite | BGR | S55L16 C | Füsseberg Mine, Siegerland, Germany | strongly altered | C1_Chalcopyrite_BG R-S55L16-C [5x5 AVG] | S: 34,941; Fe: 30,091; Cu: 34,094 |
| C2_Chalcopyrite | BGR | S115R12 | Erzgebirge, Slovakia | slightly altered | C2_Chalcopyrite_BG R-S115R12 [5x5 AVG] | S: 34,903; Fe: 30,068; Cu: 33,95 |
| C3_Chalcopyrite | BGR | S131L5 C | Henderson Mine, Clear Creek Country, USA | tarnished | C3_Chalcopyrite_BG R-S131L5-C [5x5 AVG] | S: 35,039; Fe: 30,106; Cu: 33,965 |
| C4_Chalcopyrite | UP | 7534 | Cornwall, England, GB | slightly altered | C4_Chalcopyrite_UP-7534 [5x5 AVG] | S: 35,007; Fe: 30,156; Cu: 34,044 |
| C5_Chalcopyrite | UP | 7526 | Clausthal, Harz, Germany | altered | C5_Chalcopyrite_UP-7526 [5x5 AVG] | S: 35,053; Fe: 30,007; Cu: 34,177 |
| K1_Copper | UP | 600-1 | Furnace, Lübeck, Germany | slightly altered | K1_Copper_UP-600-1 [5x5 AVG] | Cu: 98,577 |

**Table 11: Sample names, collection, original sample name, locality, alteration, mineral formula, spectral library filenames and geochemical composition of the copper-bearing silicates, carbonates and sulphates.**

| Sample name | Collec-tion | Original sample name | Sample locality | Visible alteration | Spectra name | Geochemical composition (EPMA mean, n=3, wt%) |
|---|---|---|---|---|---|---|
| A1_Azurite | UP | 2458 | Cheroy near Lyon, France | altered, nodular | A1_Azurite_UP-2458 [5x5 AVG] | CuO: 65,344; HgO: 0,091 |
| A2_Azurite | UP | 2437 | Tsumeb near Otavi, Namibia | altered | A2_Azurite_UP-2437 [5x5 AVG] | CuO: 65,194 |
| A3_Azurite | BGR | S101L7 | Cornberg by Fulda, Germany | strongly altered | A3_Azurite_BGR-S101L7 [5x5 AVG] | CuO: 63,87; $SO_3$: 0,127; FeO: 0,179 |
| B1_Brochantite | BGR | S115R3 | Altenberg, Slovakia | slightly altered, powdered | B1_Brochantite_BGR-S115R3 [5x5 AVG] | $Al_2O_3$: 0,18; $SiO_2$: 0,069; SO3: 16,262; CuO: 80,334 |
| F1_Unknown | BGR | S115R14 | Kotterbach near Witkow, Poland | slightly altered | F1_Unknown_BGR-S115R14 [5x5 AVG] | $SiO_2$: 2,588; FeO: 69,042; CuO: 0,25; $SO_3$: 0,161; MnO 0,292 |
| L1_Linarite | UP | 9542 | Unknown location | slightly altered, acicular | L1_Linarite_UP-9542 [5x5 AVG] | $SO_3$: 64,18; CuO: 24,184; HgO: 0,439 |
| M1_Malachite | BGR | S134R8 | L'Etoile du Congo Mine, Katanga, Kongo | altered, nodular | M1_Malachite_BGR-S134R8 [5x5 AVG] | CuO: 67,609 |
| M2_Malachite | BGR | S131L5 M | Henderson Mine, Clear Creek Country, USA | strongly altered | M2_Malachite_BGR - S131L5-M [5x5 AVG] | CuO: 66,917 |
| M3_ Malachite | BGR | S131R4 | Tsumeb near Otavi, Namibia | altered | M3_Malachite_BGR-S131R4 [5x5 AVG] | CuO: 65,176; $SO_3$: 0,458 |
| M4_ Malachite | BGR | S132L2 | Ogonja Mine in Okahandja, Namibia | strongly altered | M4_Malachite_BGR-S132L2 [5x5 AVG] | CuO: 67,051 |
| M5_ Malachite | BGR | S55L16 M | Siegen, Germany | slightly altered, acicular | M5_Malachite_BGR-S55L16-M [5x5 AVG] | CuO: 67,885 |
| P1_Plancheite | UP | Oberhä | Jordan | slightly altered | P1_Plancheite_UP-Oberhä [5x5 AVG] | $Al_2O_3$: 2,951; $SiO_2$: 42,079; CuO: 51,782; $SO_3$: 0,061; MnO: 0,243 |
| P2_Plancheite | UP | Oberhä2 | Jordan | slightly altered | P2_Plancheite_UP-Oberhä2 [5x5 AVG] | $Al_2O_3$: 3,727; $SiO_2$: 44,12; CuO: 48,902; $SO_3$: 0,282; MnO: 0,247 |
| P3_Plancheite | UP | Oberhä3 | Jordan | slightly altered | P3_Plancheite_UP-Oberhä3 [5x5 AVG] | $Al_2O_3$: 2,74; $SiO_2$: 43,25; CuO: 51,37; $SO_3$: 0,266; MnO: 0,085 |

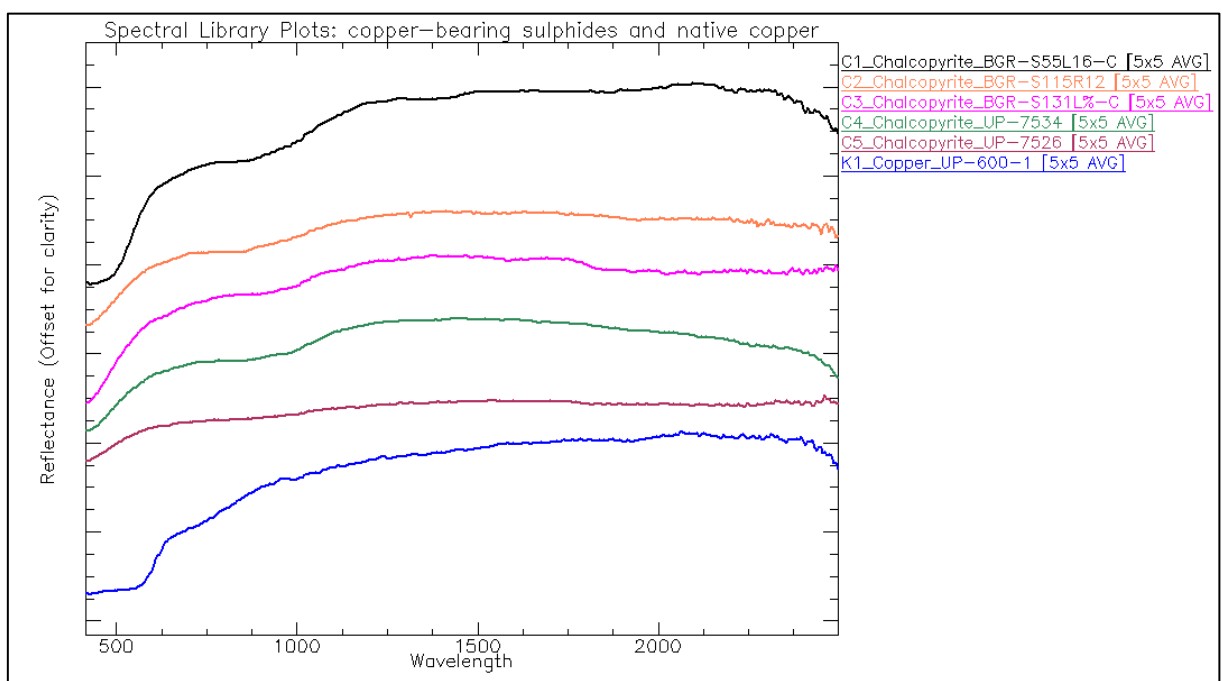

**Figure 10: Spectral library plot of the copper-bearing minerals – sulphides and native copper.**

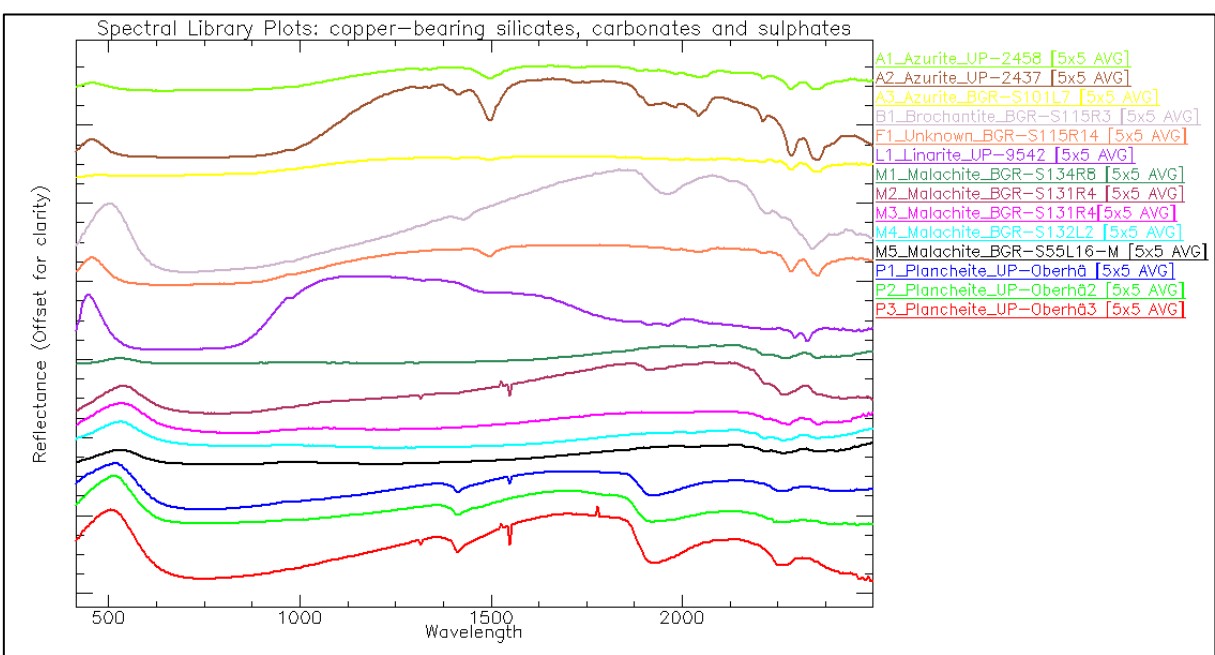

**Figure 11: Spectral library plot of the copper-bearing minerals – silicates, carbonates and sulphates.**



## 4.3 Apliki mine samples

**Table 12: Sample names, spectral library filenames, description and mineralogy of Apliki mine sample collection**

| Sample ID "Spectra name" | Description | Mineralogy based on qualitative XRD analysis (in no particular order) from Koerting (2021) |
|---|---|---|
| *Apl1_A_1a*, **"Apl1_A_1a [5x5 AVG]"** | *Grey green "fresh" surface* | Not available |
| *Apl1_A_1b,* **"Apl1_A_1b [5x5 AVG]"** | *Hematite coloured weathering crust* | Andesine (anorthic), Quartz, Magnetite, Montmorillonite |
| *Apl1_A_1d,* **"Apl1_A_1d [5x5 AVG]"** | *"Fresh", dark green weathering crust* | Anorthite, Magnetite, Diopside, Quartz, Montmorillonite |
| *Apl1_A_1e,* **"Apl1_A_1e [5x5 AVG]"** | *Yellow-ish orange weathering crust* | Magnetite, Quartz, Montmorillonite, Diopside, Anorthite |
| *Apl1_A_1f,* **"Apl1_A_1f [5x5 AVG]"** | *"Soil formation", gravel* | Magnetite, Anorthite, Quartz, Montmorillonite, Pyrite |
| *Apl1_A_2a,* **"Apl1_A_2a [5x5 AVG]"** | *Waste, soil* | Goethite, Quartz, Clinochlore, Jarosite-Natrojarosite, Andesine, Gypsum |
| *Apl1_A_3a,* **"Apl1_A_3a [5x5 AVG]"** | *Yellow-ish weathered, soil* | Andesine (anorthic), Quartz, Gypsum, Clinochlore, Jarosite, Montmorillonite |
| *Apl1_A_3b,* **Apl1_A_3b [5x5 AVG]** | *Brown-ish weathered, soil* | Quartz, Andesine, Clinochlore, Gypsum, Jarosite, Montmorillonite |
| Apl1_A_4a, **"Apl1_A_4a [5x5 AVG]"** | *White, small grained gravel,* | Gypsum, Quartz, Clinochlore, Rozenite |
| Apl1_A_4b, **"Apl1_A_4b [5x5 AVG]"** | *Grey, small grained gravel* | Quartz, Clinochlore, Andesine, Gypsum, Montmorillonite |
| Apl1_A_4c, **"Apl1_A_4c [5x5 AVG]"** | *Grey-green, weathering crust* | Quartz, Clinochlore |
| Apl1_A_5a, **"Apl1_A_5a [5x5 AVG]"** | *Grey-medium, weathering crust* | Gypsum, Quartz, Clinochlore |
| Apl1_A_5b, **"Apl1_A_15b [5x5 AVG]"** | *Grey-dark, weathering crust* | Gypsum, Quartz, Clinochlore |
| Apl1_A_5c, **"Apl1_A_5c [5x5 AVG]"** | *Grey-light, weathering crust* | Quartz, Gypsum, Clinochlore, Goethite, Hexahydrite |
| Apl1_A_6a, **"Apl1_A_6a [5x5 AVG]"** | *Red-ish brown, soil, gravel* | Quartz, Pyrite, Analcime, Goethite, Montmorillonite, Clinochlore, Anorthite |
| Apl1_A_6b, **"Apl1_A_6b [5x5 AVG]"** | *Red-ish brown, soil* | Anorthite, Quartz, Magnetite, Diopside, Montmorillonite, Gypsum, Goethite |
| Apl1_A_6c, **"Apl1_A_6c [5x5 AVG]"** | *Red-ish brown* | Quartz, Clinochlore, Analcime, Gypsum, Calcite, Jarosite, Pyrite, Montmorillonite |
| Apl1_A_6d, **"Apl1_A_6d [5x5 AVG]"** | *Red-ish brown, soil* | Quartz, Pyrite, Anorthite, Analcime, Clinochlore, Montmorillonite |
| Apl1_A_7d, **"Apl1_A_7d [5x5 AVG]"** | *Grey, crust unstable* | Quartz, Hexahydrite, Clinochlore, Gypsum, Pyrite |
| Apl1_A_7d_Hem, **"Apl1_A_7d_Hem [5x5 AVG]"** | *Red, hematite* | Pyrite, Hematite, Quartz, Gypsum, Clinochlore |
| Apl1_A_7e, **"Apl1_A_7e [5x5 AVG]"** | *Blue crystal* | Rozenite, Goethite, Quartz, Apjohnite, Ferrohexahydrite |
| Apl1_A_8a, **"Apl1_A_8a [5x5 AVG]"** | *Grey, small grained gravel* | Quartz, Clinochlore, Pyrite, Ajoite |
| Apl1_A_8b, **"Apl1_A_8b [5x5 AVG]"** | *Grey, small grained gravel* | Quartz, Clinochlore, Pyrite, Ajoite |
| Apl1_A_8c, **"Apl1_A_8c [5x5 AVG]"** | *Grey, soil-ish,* | Quartz, Clinochlore, Pyrite, Ajoite |
| Apl1_A_9a, **"Apl1_A_9a [5x5 AVG]"** | *Light green weathering crust* | Quartz, Clinochlore (Mn), Clinochlore |
| Apl1_A_9b, **"Apl1_A_9b [5x5 AVG]"** | *Hematite vein* | Quartz, Clinochlore, Pyrite, Hematite |
| Apl1_A_10a, **"Apl1_A_10a [5x5 AVG]"** | *White with pink, weathering crust* | Clinochlore, Hematite, Quartz |

| Sample ID "Spectra name" | Description | Mineralogy based on qualitative XRD analysis (in no particular order) from Koerting (2021) |
|---|---|---|
| Apl1_A_10b, "**Apl1_A_10b [5x5 AVG]**" | *White with purple, weathering crust* | Quartz, Clinochlore |
| Apl1_A_10c, "**Apl1_A_10c [5x5 AVG]**" | *Green-ish veins* | Quartz, Clinochlore |
| Apl1_A_10d, "**Apl1_A_10d [5x5 AVG]**" | *White evaporitic crust* | Quartz, Clinochlore, Pyrite |
| Apl1_A_11a, "**Apl1_A_11a [5x5 AVG]**" | *Grey, weathering crust* | Quartz, Clinochlore, Gypsum, Bassanite |
| Apl1_A_11b, "**Apl1_A_11b [5x5 AVG]**" | *Green, weathering crust* | Quartz, Clinochlore, Sphalerite |
| Apl1_A_13a, "**Apl1_A_13a [5x5 AVG]**" | *Red, rock* | Andesine, Quartz, Magnetite, Montmorillonite-Chlorite, Diopside |
| Apl1_A_13b, "**Apl1_A_13b [5x5 AVG]**" | *Red, gravel, weathered hillside rock* | Clinochlore, Quartz, Montmorillonite |
| Apl1_A_15a, "**Apl1_A_15a [5x5 AVG]**" | *Dark blue crystalline crust* | Quartz (82.6%), Pyrite (7.5%), Chalcopyrite (0.8%), Pentahydrate (cuprian) (9.1%) |
| Apl1_A_15b, "**Apl1_A_15b [5x5 AVG]**" | *Light blue rock and blue crust* | Quartz (86.1%), Pyrite (4.5%), Pentahydrate (cuprian) (7.1%), Covellite (2.4%) |
| Apl1_A_15c, "**Apl1_A_15c [5x5 AVG]**" | *Black pyrite* | Covellite (18.9%), Quartz (39.9%), Chalcanthite (21.8%), Pyrite (20.0%) |

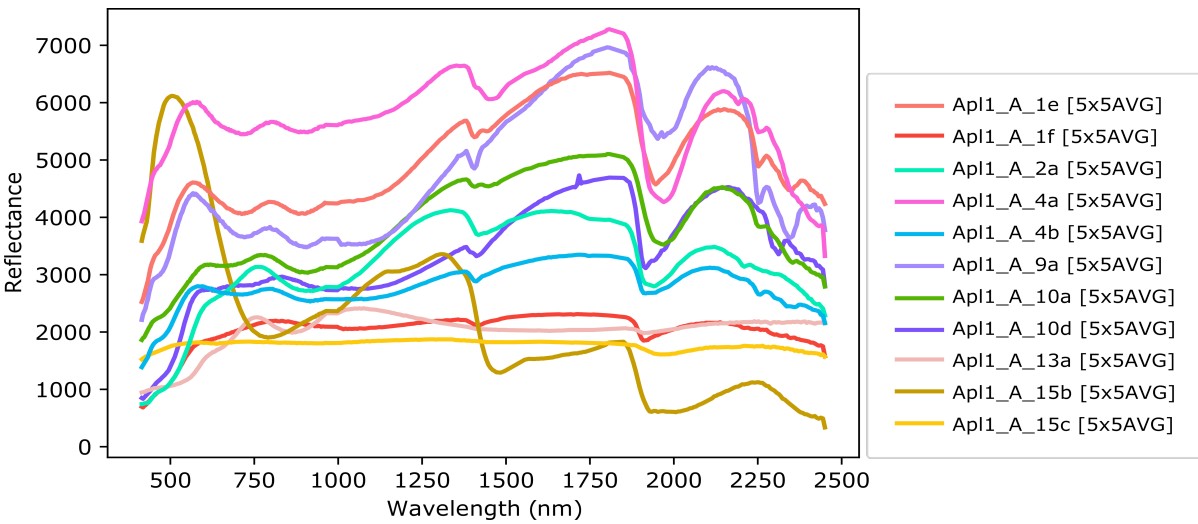

**Fig. 12: Spectral library of the 37 different Apliki mine samples, spectra stacked with offset.**

**Figure 13: Detailed, unstacked view of a selection of spectra. Reflectance scaled from 0-10000, modified from** (Koerting, 2021)**.**

## 5. Validation and Discussion


Technical validation of the results in terms of sample material properties, systematic errors and variation of measurements (experimental error) are given below.

### 5.1 Sample Material Properties

The REO powders were certified to contain at least 99.9% of the corresponding REO. The certificates are listed in (Koerting et al., 2019a). The REE mineral samples were geochemically analysed using the Thermo Niton XL3t (Fisher Scientific, 2002) device. The resulting element concentrations and the measurement error ($2\sigma$) are provided in Koerting et al. (2019a). The validation for the copper-bearing minerals can be found in Koellner et al. (2019) and the Apliki mine sample validation, analyzed by BVM, can be found in Koerting et al. (2019b).


### 5.2 Systematic Errors of hyperspectral data acquisition

Systematic errors are discussed based on instrument drift, calibration and optimization of measurements. Initializing a warm-up phase of optical components, detectors and lamps, reduced influences due to instrument drift. Additionally, laboratory conditions were monitored to ensure a stable temperature and humidity. The HySpex cameras and the reference standards are
factory calibrated once per year. Measurements used for the final reflectance spectral library were collected within one calibration time span to ensure equal acquisition conditions. For HySpex, averaging multiple measurements minimizes variations in the data. An average (median) of 500 to 800 pixels reflectance spectrum was taken for the HySpex REE and REO reflectance spectra. This number relates to the maximum number of non-disturbed pixels per sample region of interest (e.g., pixels that were not shadowed from the sample holder side walls, etc.). For the copper-bearing minerals and the Apliki mine
powders a 5x5 average pixel window was chosen over the area of interest. For these samples using a smaller pixel number for the average was necessary as the sampling of the copper-bearing minerals for geochemical validation occurred over a small area of the sample and the Apliki mine powder tablets were too small to ensure a larger homogenous area.

### 5.3 Measurements variation

Variations of measurements were not only based on instrument calibrations or drift. They can also occur due to the detector geometry or geochemical properties of the minerals. These variations may appear as a shift of the peak positions of the absorption bands. This means, different hyperspectral sensors will show variations in the spectrum of the same material. By only using one set of hyperspectral sensors, the HySpex VNIR and SWIR, these shifts will not appear in our data sets. They might show when comparing our reflectance spectra of a material with reflectance spectra taken from a different instrument.
For the copper-bearing minerals, the sample reflectance spectra also differ when comparing different samples of the same mineral species (e.g. "Malachite") to each other. The spectral signal differs, for example, due to changes in geochemistry,

physical appearance e.g. crystallization and degree of weathering (Clark, 1999; Hunt, 1982; Hunt and Ashley, 1979). To avoid measurement variations caused by different sensors imaging data from the same sensors as the spectral library should to be used for the analysis. An example for an application can be using the here provided spectral library of the Apliki mine samples

for an analysis of the HySpex hyperspectral imaging data of the Apliki mine face to be published in 2021 (Koerting et al., 2021).

### 5.4 XL3t systematic errors

The XL3t is internally calibrated and provides an internal warm-up phase to guarantee stable measurement conditions. Unlike

the spectrometer measurements, experimental error was only provided for the XL3t. In order to reduce the experimental error, a long duration measurement time of 120 seconds was set. The XL3t collects the emitted radiation from the sample using four different filters. While the sample was irradiated, each filter measures counts per second within a time span of 30 seconds. Next, the average counts per second were internally transformed to ppm. The irradiation of, in total, 120 seconds per sample was empirically tested to enable short measurement duration in combination with the lowest achievable standard deviation of

concentration level.

### 6. Data Availability

The spectral libraries are published under the Creative Commons Attribution International 4.0 Licence (CC BY 4.0) via GFZ Data Services. Due to the different types of samples, we present the following three data publications: (1) Mineral reflectance

of 29 rare-earth minerals and rare-earth oxide powders including niobium- and tantalum-oxide powder, V. 2.0 GFZ Data Services, http://doi.org/10.5880/GFZ.1.4.2019.004 (Koerting et al., 2019a); (2) Mineral reflectance spectra and chemistry of 20 copper-bearing minerals, V. 2.0 GFZ Data Services, http://doi.org/10.5880/GFZ.1.4.2019.003 (Koellner et al., 2019) and (3) Mineral reflectance spectra and chemistry of 37 copper-bearing surface samples from Apliki copper-gold-pyrite mine in the Republic of Cyprus, V. 2.0 GFZ Data Services,  http://doi.org/10.5880/GFZ.1.4.2019.005 (Koerting et al., 2019b).

### 7. Sample Availability

The samples provided by the BGR are available through the collection of the BGR Spandau by their sample and collection name in the technical report (https://www.gewis.bgr.de). The samples provided by the GFZ and UP belong to projects and have to be requested separately.

## 8. Appendix

**Table A1: List of less commonly known terms and their abbreviations used throughout the paper**

| Terms | Abbreviation | Description |
|-------|--------------|-------------|
| Abbreviation | REE | Rare Earth Element |
| | REO | Rare Earth Oxide |
| | REMin | Rare Earth Element-bearing Mineral |
| | VNIR | Visible light and near infrared |
| | SWIR | Short wave infrared |
| | XRF | X-Ray fluorescence |
| | EnMAP | Environmental Mapping and Analysis Program: future earth observation satellite mission (www.enmap.org)[1] |
| | CCRSS-A | China Commercial Remote Sensing Satellite System: future earth observation satellite mission |
| | HISUI | Hyperspectral Imager Suite: future earth observation satellite mission |
| Instruments | HySpex VNIR-1600 | HySpex push broom spectrometer, VNIR camera |
| | HySpex SWIR-320m-e | HySpex push broom spectrometer, SWIR camera |
| | HySpex ground | HySpex operational software for laboratory and near-field application |
| | HySpex rad | HySpex calibration software to transform raw DN into radiance data |
| | Thermo Scientific Niton XL3t | Thermo Scientific Inc. X-Ray fluorescence analyzer (NITON TM XL3t) |
| | NDTr | Thermo Scientific Inc. NITON TM operational software |
| | JEOL JXA-8200 | Electron probe microanalyzer (EPMA) |

| Terms | Abbreviation | Description |
|---|---|---|
| | JEOL JSM-6510 | Scanning electron microscope (SEM) |
| | Oxford Instruments INCAx-act | Energy dispersive X-ray spectrometer (EDS) |
| Registered brands, Copyrights and/ or other protected terms | REacton® | Series of rare earth metals and compounds |
| | Alfa Aesar | Manufacturer and supplier of chemicals for research and development (today: "Thermo Scientific Inc.") |
| | Gunnar Färber Minerals | Supplier of mineral specimen |
| | REEMAP | Rare Earth Element MAPping: Research project for the development of a modular multi-sensor processing chain for modern imaging spectrometers to detect REEs |
| | Smithonian Institution | Smithsonian Institution Department of Mineral Sciences, reference material from the Smithonian Microbeam Standards |
| | Astimex Standards Ltd. | Astimex produces standards suitable for electron probe and scanning electron miscroscop X-ray analysis. |
| | BVM | Bureau Veritas Minerals is an industry leader in the analysis of minerals for the Exploration and Mining industries. BVM is a service-provider company that provides mineral preparation and laboratory testing services. |
| Research and federal institutes | BGR | Federal Institute for Geosciences and Natural Resources |
| | GSD | Geological Survey Department, Ministry of Agriculture, Rural Development and Environment, Republic of Cyprus |
| | UP | University of Potsdam |
| | GFZ | German Research Centre for Geosciences |
| Registered Trademarks | Excel™ | Microsoft Excel™ |

## 9. Author contributions

*Apliki mine and copper-bearing minerals:* Friederike Koerting designed the Apliki sample related study, performed and the measurements of the Apliki samples and wrote the manuscript. Nicole Koellner designed the copper sample study, supervised the measurements and performed the geochemical analysis at the University of Potsdam. Christian Mielke and Agnieszka Kuras prepared parts of the spectral libraries. *REE minerals and REOs:* Nina K. Boesche designed the REE study, performed some measurements, and supervised the REE measurements. Sabrina Herrmann prepared the samples and conducted most of the measurements. Christian Rogass developed and applied the HySpex post-processing chain. Christian Mielke and Kirsten Elger helped revising the manuscript. Uwe Altenberger supervised the studies and gave valuable comments on the manuscript.

## 10. Competing interests

The authors declare no conflict of interest

## 11. Acknowledgements

We would like to thank the Helmholtz Centre Potsdam GFZ German Research Centre for Geosciences for providing the infrastructure and personnel support to conduct our research. Our gratitude also goes to the German Federal Ministry of Education and Research and the r4 subsidy program for innovative technologies for resource efficiency, which supported the REEMAP scientific project. We also thank the DLR Space Administration and the German Federal Ministry for Economic Affairs and Energy for the financial support based on a decision by the German Bundestag in the frame of the EnMAP scientific preparation program (Contract No. 50EE1256). We also want to express our gratitude to Seltenerdmetalle24, in person Manuel Schultz, for his friendly service when providing laboratory standards and negative control sample holder.  Thanks to the support by the GSD we were able to conduct a study and sample in the Republic of Cyprus and our thanks goes to our colleagues there for their help and directions in the unknown terrain. All the work in the Republic of Cyprus was conducted under the "Permit to conduct a Geological Survey, Ref. No. 02.13.005.002.005.022" from the 19[th] of March 2018, granted by the Geological Survey Department, Ministry of Agriculture, Rural Development and Environment (GSD) and the Director Dr. Costas Constantinou. After the termination of the permit, a Memorandum of Understanding (MoU) and Framework for cooperation in the area of geo-science between the GSD and the GFZ was agreed upon in March 2019, the publication of the Apliki mine surface data is associated to this MoU. Constantin Hildebrand and Friederike Klos prepared parts of the spectral libraries for the data publications and contributed insight into the spectral interpretation. Pia Brinkman prepared the Apliki sample powder tablets. Marcel Horning performed most of the measurements on the copper samples and prepared the spectral copper library during his B.Sc. thesis. We thank our colleagues for their input and insights.

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
