# Peer review of "A solar optical hyperspectral library of rare earth-bearing minerals, rare earth oxide powders, copper-bearing minerals and Apliki mine surface samples"

_Earth System Science Data, 2019_

## Referee Comment (RC1) · Jeanne Percival (Referee) · 19 Feb 2020

**A solar optical hyperspectral library of rare earth-bearing minerals, rare earth oxides, copper-bearing minerals and Apliki mine surface samples.**

Koerting, Koellner, Kuras, Boesche, Rogass, Mielke, Elger and Altenberger

Earth System Science Data-Copernicus

**General Comments**

This paper provides an overview of several spectral libraries developed on REE-bearing oxides and minerals as well as copper-bearing minerals and rock and soil samples. The use of these VNIR spectra in remote sensing is valuable for the exploration industry, and can be applied in environmental type studies. In addition, the authors have indicated they provided geochemical validation of the samples by various means. The paper is succinctly written and is supported by a supplement as well as several datasets.

There is room for improvement in all the documents. Access to the Potsdam University datasets was not easy, and without a password for logging in as "anonymous" I did not obtain access to the geochemical data for review (except in one case). However, I was able to access the text files of the spectral data, and was able to re-plot them. I have no issues at all with the spectral data. One set was not plotted in the datasheets (REO powders) and this should be added, similar to the REE-bearing minerals. The authors should also explain exactly what they mean by geochemical validation-to show the composition of the samples, or to show a relationship between composition and absorption minima?

Suggest authors to visit doi.org/10.4095/315690 for a look at a Spectral Library on REE-, U-, Th- and Nb-bearing minerals by Percival et al., 2019.

**Specific Comments and Editorial Suggestions**

*Main Text*

Abstract
Line 18: Not sure I would call this an "extensive" collection of minerals
Line 23: "openly available" – I could not access the geochem datasets as I had to have an account at Potsdam University or set up an account. I had no desire to do this. These data sets should be available as the spectral library was made available, or somewhere there should be a note as to how to access the files as a guest.

Introduction
Line 37-38: Suggest changing this sentence, as the Spectral Library is not based on the geochemical validation-it is the hyperspectral measurements. Suggest: "In this study, the spectra were collected under standardized laboratory or field conditions and include geochemical validation of the sample materials.
Line 39: add a comma after "example"

Line 40: "elemental" from a hyperspectral viewpoint, I would see Fe as the main element observed (i.e., in gossans), as REE-bearing minerals tend to be in trace amounts, very small, hence seeing the NIR signature may be masked by other minerals such as clays or carbonates etc.
Line 40: "artificial surfaces"-what does this mean? Cut or polished surfaces?
Line 41: use "mapping" singular form
Line 46: should references be in chronological order from oldest to youngest?
Line 48: Turner et al 2014a is out of order in the ref list.
Line 49: Delete "also" between the verbs, maybe start sentence with "Also, …."
Line 50: add "hand" in front of "samples" [hand samples]
Line 51: core-use singular form; delete "melt evolution". Melt evolution is something you may study in igneous petrology, and would need very detailed mineral chemistry to "visualize'
Line 68: add "in" after "shown"
Line 69: Start a new paragraph here with "we are…"
The information provided in lines 69-80 becomes repeated again below. Try to limit this, especially in such a short paper.
Line 84-88: Delete, as this is repeating what noted earlier. Add the last sentence to the end of the previous paragraph. "The spectral libraries…." after the url address.

Materials
Line 97: delete "here presented"; use "comprise"
Line 97-98: repeated from the introduction, can probably deleted this first sentence and begin with: "The REE sample material includes 16 REO…."
Line 99: change "comprises" to "includes', deleted "used'
Line 100: Were these purchased from Alfa Aesar? If so, then say purchased; if they were "donated" say that, but not "received".
Line 101-102: "…of the specified REE based on their concentration certificates, which can be found…."Line 104: "specimens"; "denotation" is not correct here in this usage-do you mean "notation" or "abbreviation" or identification"?
Line 106: "denotation" same issue-named or noted? Why should the reader consult the data description to validate the name? I could not access the XRF results- I think this should be re-worded.
Line 103: where is the appendix and Tables A2 and A3? Do you mean Supplement and Table S2 and S3?
Line 109: ditto Tables A4-A5
Line 111: "measurement campaign" to collect samples, map, do field work or just measure using the cameras? Can you say "field campaign"?
Line 115: Table A6?; "overview of sample…"
Line 116: Correct all references to appendix if it is the "supplement"

Sample preparation and spectra collection
Line 122: "depended"
Line 123: "number of pixels"
Line 124: "compiled" rather than "collected"? Add a comma after "Thereby, …"
Line 125: "spectral homogeneity"
Line 129: comma after "measurements"
Line 132: "pixels"

Line 135: use "example" not "exemplary"; copper-bearing minerals [need hyphen]
Line 136: Table A5-Table S5 in the supplementary information?
Line 140: use "example" not "exemplary"; change "non-existent" to "lack of"
Line 145: Change: "…powdered so that > 85% of the sample was < 75 µm"
Line 147: "tablet's metal frame" [need apostrophe]

HySpex data recording
Line 156: why is the beginning bolded?
Line 157: combine sentences by deleting ".the two cameras are"
Line 162: do you mean "sleigh"? sleight means cunning-check definition.
Line 164: ditto
Line 167-169: need a reference added in here
Line 179: change "exemplary: to "example"

Thermo Niton XL3t (XRF)
 Line 227: Bring up to the paragraph before; this is a field-portable or hand-held instrument, not "mobile"
Line 227-233: This is repeated in the data set paper, why include this detail here. If this is the preferred place, then shorten that in the data set paper. Principles of XRF analysis are not really needed, can refer to a paper on this subject.
Line 233: place praseodymium before neodymium as this is the order in the Periodic Table.
Line 236: use "made of plastic with a plastic foil…."
Line 237: "built-in"
Line 238: "not "used software" but "software used"
Line 239: Delete "the" in front of "mining and exploration" and delete "mode" [already noted in Line 238]
Line 240: "identified" do you mean "attributed"?

Scanning electron microscope (SEM) and Electron probe microanalyzer (EPMA)
Note that EPMA should be used rather than EMPA, so change terms accordingly in the text and in tables
Start this section with Lines 259-266: SEM should be described first.
Line 265: "qualified"-do you mean "quantified"? Change next line to "Based on previous results, divergences of up to 5 wt% can be expected, which for quantitative analysis is considered acceptable".
Line 266: Change "of a total to" to "and normalized to 100 wt%" [this is problematic]
Line 250: change electron microprobe (EPMA) to electron probe microanalyzer (EPMA)
Line253: delete "respectively"; List background count times on EPMA, especially for the trace elements.
Line 254: Measuring F in fluorapatite is problematic using EPMA. At a minimum, the Area-Peak Factors method should be used otherwise result is usually spuriously-high values.
Line 256: EPMA
Line 257: Give reference for the data reduction routine used.
Line 268-274: delete, and any additional information incorporate with the EPMA section.
Line 271: "Analyses were calibrated using natural…."
Line 276: "…the full SEM and EPMA data files…"

Line 282: "EPMA";
Line 284: ditto in Table caption. Whys is Table abbreviated to Tab? Add "column" after "comment"

Apliki mine sample analysis at Bureau Veritas Minerals
Do not need name of company in the section subtitle. Rather than the information you have provided, I suggest you write out what the analyses are-and refer to the packages via their website. For example " geochemical analyses of the Apliki samples were completed by Bureau de Veritas (location) using their standard packages (website). Samples were pulverized and then digested using aqua regia and major, minor and trace elements were determined using ICP-MS" etc.
Line 290: why was one sample analysed using "aquatic" What is so different about it-this is worth explaining. The code info is better placed in the data sheets information.

Validation and discussion
Line 306: Delete "This section will discuss the" and start with "Technical…."
Line 307: Add "are given below" after "error)"
Line 310-314: Only certificate info provided as validation for the REO powders
For REE minerals, there is no table comparing the results from the XRF vs. EPMA and indicating if the results are actually comparable. For the geochem analyses by BVM there is no information on how good the data is, no comparison of duplicates, no CRM's, unless this is also held in the University site.
Line 321: use "measurements" twice, maybe state "averaging multiple measurements minimizes variations in the data"?
Line 327: change "bigger' to "larger"
Line 335: Change "denotation" to "species"
Line 336: add commas before and after "for example"; use "degree of weathering" and delelet "grade"
Line 337: provide information on how to obtain easy access to this geochemical data.

Data availability
Whys is this section bolded?
Line 349: Delete "here presented"
Line 350: change "are presenting" to "present"

Sample availability

Line 357: Table S5? Note that "Table" should be capitalized throughout the paper-inconsistent.

Appendices
Line 365: So here are "appendices' but only one Table {do not abbreviate Table]

References
Turner et al, 2014a is out of order, Turner 2015 should be first (single author). Leave a space between Tong et al and the turner reference.

Figures
Fig. 1: This is very good and provides a good image of what your paper is about. Suggest that the spectral pattern for chalcopyrite be a darker colour, cannot see the yellow; also the malachite should be a continuous line like the other three.

Tables
Table 1 is not really necessary-you can summarize that info in the text.

Suggest combining Tables 2, 3 and 4 into 1 as many parameters are common. So you can have more columns relative to each of the different mineral groups.
Table 5: The header "Concentration level determination" does not relate to what you have placed in the rows. What are they? All you do is re-reference the data sets. Likely this Table is not critical to the paper.
Table 6: What about adding in the probable interferences for analysing the LREE with a hand-held XRF instrument. This provides very little information to the reader.
Table 8 and 9: Is this really important? You have not noted any samples numbers in the text so there is information for the reader in table 8. This should be in an Appendix if deemed critical. Describe the analyses and how they are done –that is more important.
Table A1: rather than "not commonly" use "less commonly" in caption. "List of less commonly known terms and their abbreviations used throughout the paper."
MS-Excel™ I believe needs to be added

**Mineral spectra and chemistry of 32-rare-earth minerals and rare-earth oxides including niobium- and tantalum-oxide**

Koerting, Hermann, Boesche, Rogass, Mielke, Koellner and Altenberger

Earth System Science Data-Copernicus

**General Comments**

See comments for main paper
Authors did not plot the spectra for the REO pwders-why?

**Editorial Suggestions**

Citation
References: Koerting et al. 2019, Koerting et al. 2019a, and of course there was a Koerting et al. 2019b in the paper. For the data sheets, maybe use 2019 a and b? Maybe editor can resolve this.

Abstract
Line 5-6: "…tantalum-oxides commonly associated with REEs".
Line 6: change "bigger" to "larger"

Samples
Line 1: use Table 1 [cap T]
Line 3: Were these REO powders purchased or were they donated? Change "received"
Line 4-6: "…REE and were accompanied by concentration certificates (Table 4). Delete other "delivered together" and "these…..description".
Line 8: "…online trader of minerals specimens"; use "notation" rather than "denotation", or "name" or "identification"
Line 9: change "data" to "results"- Why should the reader consult the data description to validate the name?
Line 10: ditto on "denotation"
Line 11: Where is Table A2 and A3-are these S2 and S3? Should be In Supplement rather than Appendix? Last sentence: "All of the samples were analysed as part of the MSc…and PhD…."

Table 1: Gadolinite [missing e]
Remove extra periods
Note; Ideal Formula

Hyperspectral measurements
Line 1: Why is the first part bolded?
Line 2: combine this short sentence with previous one (as in the paper) "…two line scanning cameras mounted in parallel"
Line 3: Add "region" after (…2500 nm)"
Line 6: "sleigh"

Line 9: ditto

Geochemical measurements

Line 1: "corresponding concentration level determination"-what does this mean? Do you mean geochemical method? There is nothing in Table 3 about concentrations.

Use EPMA

Table 4: Curious, how did these two co-authors modify the certificates of purity for the REO powders?

(lines numbered after Table 4)

Line 2: electron probe microanalyser (EPMA)

Line 6: use "hand-held" or "field portable" rather than "mobile"

Line 13: place Pr before Nd [order in P.T.]

Table 6: I believe this is Table 5

(lines numbered after Table 5)

Line 1: change "additionally" to "also"

Line 2: EPMA [I realize on the website it is listed as EMPA-but most now use the term EPMA]

Line 2: comments on probe info-see editorial suggestions for the "paper"

8.1 Spectra: ENVI Spectral Library

Relatively easy to see the spectral data and plot each sample out individually. For the REE minerals, can identify them by the numerous deep reflections, especially in the NIR. What is really needed, is to identify which REE corresponds to which reflection-as in Turner's 2015 thesis (and published papers 2014). This is the test for validation.

Why did you not plot the REO powders in the same format in this section?

8.2 Geochemical Analyses

I do not see REO analyses in any table, certainly not in Table 5.

Note Excel™ should probably include the trademark symbol [throughout the paper]

I could not access geochemical data by XRF nor EPMA.

Fig. 3 caption: use "example" rather than "exemplary" [wrong meaning]

Note: sometimes able is not capitalized, sometimes Figure is shortened to Fig. under the figure. Be consistent.

References

Sometimes the date is at the end, other times after the authors.

**Mineral spectra and chemistry of 20 copper bearing minerals**

Koellner, Koerting, Horning, Mielke and Altenberger

Earth System Science Data-Copernicus

**General Comments**

The information provided in this data description is similar to what is provided in the Supplement, especially the fact that both have the Table with photos of the samples. I believe this is the place where that should be held, and maybe the Supplement does not need to be so extensive with images.

For this set of data, I was able to open the EPMA results as well as the SEM images and EDS analyses. I assumed that the EPMA of the 25 analyses match the samples depicted in the SEM file, however that was not the case. It would be very useful in the EPMA files to indicate the sample name using the abbreviations provided (A1, A2, A3, etc.) for cross-reference. Also, as you have several samples of one mineral, averaging their mineral chemistry and calculating an actual mineral formula would be most useful. This can be also calculated for the individual samples.

With respect to the EDS results shown in the pdf. Fil, is there any reason to indicate "H2O missing"? What is more useful is to explain this in the data sheets, and emphasise that EDS analyses are normalized and WDX analyses are not.

**Editorial Suggestions**

Citation
Line 1: Change "bigger" to "larger" [ditto in abstract]

Abstract
Line 1: there are 20 samples, but only 7 or 8 different minerals (native copper, azurite, malachite, chalcopyrite, plancheite, brochantite, linarite, and an unknown). Why is the unknown still unknown? Is it possibly a new mineral? Is it crystalline or amorphous? What is the XRD trace of this mineral?
Line 4: For this group of data it was accessible.

Samples
Line 3: delete "(EDX)" and "(WDX)" [noted under methods]; in brackets should be "(SEM)" and "EPMA)"
Figure 1 caption: use "example" not "exemplary"; change "non-existent" to "lack of"
"…example of all sample scans to highlight the lack of sample preparation"
Line 5: Delete "previous" [can use "prior"]
Line 10: change "for" to "of" [area of the geochemical…]

Table 1
What is spicular? [M5 and L1]
Table 2
Add the initials used to indicate mineral name at end of caption.

Hyperspectral measurements
Line 1: why bolding used?
Line 2: combine sentences [as noted in other review docs]
Line 3: add "region" after wavelength brackets
Line 6 and 9: "sleigh"

Mineral chemistry
Line 7-8: "qualified"-do you mean "quantified"? Change next line to "Based on previous results, divergences of up to 5 wt% can be expected, which for quantitative analysis is considered acceptable".
Line 10: EPMA
Line 17: ditto
Line 18: change "validated" to "estimated".

7.1 ENVI Spectral Library
No issues with the spectral data; can open the data files. What validation has been done between the spectral signature of a samples and its geochemistry?

7.2 Detailed sample list with measured parameters
Excel™

7.3 mineral chemical analyses
You provide an example of the raw data for EPMA analyses. Why not summarize the chemistry in a table and indicate calculated formula for each mineral?
What are the ore minerals?

Q: text file with copper mineral chemistry for 25 samples-this appears top relate to the pictures in the pdf of "copper minerals chemistry"-correct?
What are the 51 copper minerals in the other text file?

Figure 6 caption. Use "example" not "exemplary"

**Mineral spectra and chemistry of 37 copper-bearing surface samples from Apliki copper-gold-pyrite mine in the Republic of Cyprus**

Korting, Rogass, Koellner, Kuras, Horning and Altenberger

Earth System Science Data-Copernicus

**General Comments**
The information provided in this data description is similar to what is provided in the main paper and the Supplement, especially the fact that both have the Table with photos of the samples. I believe this is the place where that should be held, and maybe the Supplement does not need to be so extensive with images.

For this set of data, I was not able to open the geochemistry results. I was able to open the spectral files and re-plot the spectra.

**Editorial Suggestions**

Citation
Line 1: "larger" rather than "bigger"

Abstract
Line 3: change "sampled' to "collected" and "of the" to "with the"
Line 8: ditto "bigger"

Samples
Line 1: use: "The 37 samples were collected" or "Thirty-seven samples were collected…"
Line 2: what is the "measurement campaign"? Was this a hyperspectral measurement field trip? delete "Cyprus in front of "geological"
Line 4: "pulverized so that $\geq$ 85% of the samples were < 75 µm"

Hyperspectral measurements
Line 4: "the area for the sample's spectra was…"

Table 2:
See comments for the Supplement documents
What is "soil-ish"?
Some samples are described as "white with pink"-what are these, soils, gravels or rock?

Geochemical analyses
Line 2-3: Table 3 and Table 4

Table 3 and Table 4 were also in the main paper-they are not really important here; provide a web address. I think a description of the method is more valuable to the reader. Also, need to provide accuracy and precision information, there is no mention of duplicate or CRMs being analysed. Geochem data could not be accessed.

81. ENVI spectral library file
No issues with the spectral data; can open the data files. Plotting them in one image makes most very flat. What validation has been done between the spectral signature of a samples and its geochemistry?

8.2 Extensive sample list and measured variables
Note Excel™ needs a trademark symbol

Table 6 or 7 not identified
Are these needed? Repetitive from Tables 3 and 4? Why was aquatic method used on one sample?

---

## Referee Comment (RC2) · Jeanne Percival (Referee) · 24 Feb 2020

I did note that to the editors that it might have been because of our government firewall. But it was unusual in that I could see one set of geochemical data, but the other two directed me to the university site with a login request.

---

## Short Comment (SC1) · 24 Feb 2020

Dear Jeanne Percival,

many thanks for your careful review of our manuscript. I will ony comment on some data availablitiy issues raised in your report:

(1) missing link from the data to the ESSDD paper: it is now crosslinked, please apologise the delay and thanks for informing us!

(2) data access barriers: I was very surprised reading this as GFZ Data Services does not put any regiatration or access requests to their published data. I have tried it again this morning and can directly download and view the geochemical data from aal three data publcations (copper bearing: http://doi.org/10.5880.GFZ.1.4.2019.003, REE http://doi.org/10.5880.GFZ.1.4.2019.004 and Apliki: http://doi.org/10.5880.GFZ.1.4.2019.005) without any login requests. Could you please described your access problems a little more detailed to avoind these problems in the future? I know that some firewalls are not allowing to access anonymous ftp folders, could this be the reason?

many thanks and best regards,

Kirsten Elger

(Repository manager of GFZ Data Services)

---

## Short Comment (SC2) · 25 Feb 2020

Dear Jeanne Percival,

thank you for your detailed comments on our manuscript. I started to work your suggestions into the text yesterday but I will wait for further comments on the manuscript before we start changing some of the compositional changes that you suggested. I am honestly surprised that some of the data was not accessible. Kirsten already checked and I just downloaded all of the data from the repository again. All of the data is

available at http://dataservices.gfz-potsdam.de/ and nothing should forward you to the website of the University of Potsdam or be locked by a password. Being able to access the data is the most important part here so thank you for bringing this to our attention!

Thanks again and kind regards, Friederike Körting

---

## Referee Comment (RC3) · David Turner (Referee) · 15 Jul 2020

**A solar optical hyperspectral library of rare earth-bearing minerals, rare earth oxides, copper-bearing minerals and Apliki mine surface samples.**

Koerting, Koellner, Kuras, Boesche, Rogass, Mielke, Elger and Altenberger

Earth System Science Data-Copernicus

**Manuscript Comments**

**General**

This contribution by Koerting et al includes a main paper, a supplement, and three datasets. The three datasets include description documents and multiple supporting data files.  The three datasets also have their own publication citations as GFZ Data Services products, and the reviewer was required to source the information for the ESSD paper from the GFZ publications.

While it is clear that there are linkages between the datasets, there is not linkage between all the topics. For example, the REE-related spectra match well with the Cu-related spectra, but not the Apliki site data. It is this reviewer's opinion that two ESSD papers could be written, perhaps best to match the REE and Cu mineral-focused spectra together in a single paper, and the Apliki site data by itself. That way, the target information is more consistent within the described datasets. In the present format, the four suites of data are all different in the physical nature of the samples and the (geo)chemical and mineral characterization.

Nevertheless, it is great that the raw data is being shared with the broader scientific community and that there is supporting data on the nature of the samples studied. These types of data can be important for academics, government and industry to carry out their tasks, whether that be in-field mine wall scanning or space-borne monitoring of mining activity in remote regions.

The manuscript starts with some fairly sweeping comments in the introduction that could be toned down in the context of this contribution. Formatting is not consistent, and there are simple typos and spelling mistakes that suggest the manuscript could have been reviewed again before submission (e.g., a single space before a period, abandoned commas). Sections are referred to as chapters, and appendices referred are not really appendices, but rather supplemental files that the reader must go find.

It is the reviewer's opinion that due to these observations, the manuscript should be given a re-read and edit by the authors to make it consistent with the journal structure and to address simpler non-technical issues that detract from the reading and scientific value of the data within. In order for the broader community to use any spectral library, documentation needs to be clear and there should be no ambiguity regarding data source or characterization methods. In this sense, I also recommend splitting the paper into two discrete contributions so as to reduce the potential confusion over what methods were used on which samples. Furthermore, the samples that are *rocks* and not *minerals* need to be identified as such. For example, the monazite and synchysite spectra are not what I would have

expected if these were mineral specimens. Since one of the principle uses of these spectra are as inputs to understand unknown spectra in other datasets, it is important to state their true nature.

I am happy to see this data made available to the broader community, and another pass at cleaning it up will greatly improve its usefulness!

DT.

**Specific**

Line 70. REE oxides are not minerals, though they are likely crystalline. Do you know the structure of these REE Oxides?

Line 101. Text indicates 99.9% of the specified REE, however, these samples are rare earth element oxides, and thus contain oxygen. Presumably the purity relates to not having other lanthanides, however, these are important distinctions to catch for data repositories. (clarification is given later, but as stated it still needs correction here)

Line 104. How did you validate the mineral species?

Line 108. How did you validate the mineral species?

Table 1. Why two suites for Copper bearing minerals? State why.

Line 121. "The sample preparation varied by sample type and depends on the material and the information of interest." This is a bit problematic. It is not that any one of the approaches isn't valid, but the presentation of the data as a collection should strive to have internally consistent methods.

Line ~130. You need to have example images of REMin and REO samples like you do for the Cu and mine samples

Line ~130. I'm confused. Do the REMin spectra originate from one single pixel in the middle, or an average spectrum from more pixels?

Line 134. Be sure to reference these as copper-bearing, as copper itself is a mineral… in that sense, these are not all copper mineral samples, as they are other copper-bearing minerals too (e.g., azurite).

Line 136. Text speaks about geochemical sampling, but up to this point it didn't seem like the Cu minerals were analyzed geochemically at all?

Tables 2-4. What light source did you use? Halogen?

Line 205. You should state here whether you resize the VNIR or resize the SWIR images, since the spectra are supposed to be 5 x 5 averages… this could mean quite different numbers of 'pixels' depending on whether you are upsampling or downsampling.

Table 5. I see now that you've used the SEM and EMPA for Cu minerals

Line 261. The BSE images show differences in brightness from mean atomic number of the pixel, not light and heavy elements.

Line 263. This paragraph should be combined with the previous SEM paragraph.

Section 5.2 Why do you have two distinct descriptions for the JEOL unit at Potsdam?

Line 272. Which standards? Same as above? If so, many of the Smithsonian materials are synthetic.

Line 274. Why do you speak of limitations here of an EMPA, but not of the limitations for the Niton XL3t?

Line 290. There is an aquatic sample? But the documentation states that the aquatic sample is 1a… and your index shows a rock? And Figure 3 in the main manuscript shows the sample as a powder… Does "Aquatic" mean something else? Perhaps aqua regia digestion? This needs clarity and correction.

Line 335. You need to distinguish mineral specimens/samples from mineral species

Line 338. Is Misra et al the most appropriate reference for this statement? Maybe something more classical in the geological remote sensing literature?

Line 345. 120 seconds isn't really short for a HHXRF

**Data Paper Comments**

Generally speaking, the content of the PDFs can be cleaned up and typos addressed. In this review, I have not listed them.

**Apliki**

- To confirm above points, the "aquatic" sample above is not an aquatic sample. Instead, it was analyzed by aqua regia digestion, as described in the BVM analytical notes. Why was this one sample analyzed with a different method? The manuscripts need to have this item addressed.
- It is unfortunate that your overlimit Cu samples were not re-analyzed for total Cu content, since this is a key focus for the Cu mine related samples.
- Figure 2. Maybe you could split this out into two plots, rocks and soils, so that the reader can maybe make a small assessment of the data without downloading and plotting the spectra?

**Copper Minerals**

- I see now how you've selected the small areas in the broader sample. This approach should be described in the main paper.
- Sample C3 has a typo in its name, where a "%" is used instead of "5"
  - This sample is also not just chalcopyrite, it is also bornite
- There are some samples that are not monomineralic, and this is important to state in the documentation. For example, "Malachite 1" has a whack of other minerals in the SEM image, so right away we know that the "Malachite" spectrum here is actually at least Malachite+pyroxene+quartz.
- Similarly with Azurite 2

**REE oxides, minerals and Nb-Ta minerals**

- You should include the anticipated crystal structure of these oxides. The work by White (1965?... off the top of my head) showed the importance of this variable for the resulting spectra. This is especially relevant in the context that Tb is listed as both 3+ and 4+.
- In your spectral library, ilmenite is spelled incorrectly
- In your text, synchysite is spelled incorrectly
- Similar to the copper minerals, some of these samples are not monomineralic and instead are *rocks*, and therefore the spectra need to be identified as such. For example, the images you show in the EMPA data indicate that synchysite has a bunch of phases, as does ilmenite.
- While it is not possible to do this validation with your other non-SEM samples, one can assume that this is true for at least the monazite sample, and likely your others that have multiple minerals in Table 1.
  - Your XRF data could be used to support this effort. For example, the high Sr in parisite suggests this is not only parisite and/or is another Sr-bearing REE-bearing carbonate phase.
  - Equally true about Zr in some samples
  - You need to critically evaluate the Niton data a bit more… for example the Ni values are suspect. In this sense, you need to clearly state how you're using that data to achieve your goals.
  - Similarly, it is not the voltage that preclude the analysis of other REE, but rather the fluorescence overlaps from other elements and the weakness of the L-lines that I suspect that unit is using to quantify the REEs.
  - https://www.bruker.com/fileadmin/user_upload/8-PDF-Docs/X-rayDiffraction_ElementalAnalysis/HH-XRF/Misc/Periodic_Table_and_X-ray_Energies.pdf
- "Xenotime a" is mislabeled in the XLS file
- Do you have analyses of the monazite sample?

---

## Author Response (AR1)

**Response to reviewer's comments (J. Percival)**

**Datapaper + Supplements**

**1) The authors should also explain exactly what they mean by geochemical validation-to show the composition of the samples, or to show a relationship between composition and absorption minima?**

The hyperspectral spectra are presented for different surface materials without any interpretation. The geochemical analyses are there to support the characterization of the samples and not to validate the spectra. We do not aim to present the relationship between composition and absorption minima we only aim at providing hyperspectral datasets of minerals and mixed surface materials that area also geochemically characterized.
Geochemistry and spectra of e.g. Apliki can be used separately in order to characterize the area with the provided GPS data.

**2) These data sets should be available as the spectral library was made available, or somewhere there should be a note as to how to access the files as a guest.**

The data is openly available on the GFZ data service website. There might be issues with the firewall of the user's computer.

**3) "elemental" from a hyperspectral viewpoint, I would see Fe as the main element observed (i.e., in gossans), as REE-bearing minerals tend to be in trace amounts, very small, hence seeing the NIR signature may be masked by other minerals such as clays or carbonates etc.**

This is just meant as a basic introduction for hyperspectral data uses and not tailored to outcrops or weathered surfaces. The mapping meant here, is not only for rock outcrop scan or satellite imagery but also in laboratory scans and in these scans, we do have REE mineral specimen that need to be mapped. Here, the spectrum depends on the element content and is not covered by clay etc. as if found in the field. You are right, in field imagery, the major element mapped is Fe but in laboratory scans, other elements like Nd embedded in monazite can be mapped.

**4) Line 46: should references be in chronological order from oldest to youngest? + Turner et al 2014a is out of order in the ref list.**

This is based on the template of the ESSD citation, which is in alphabetical order

**5) Fig. 1: This is very good and provides a good image of what your paper is about. Suggest that the spectral pattern for chalcopyrite be a darker colour, cannot see the yellow; also the malachite should be a continuous line like the other three.**

I changed the malachite to a continuous, the color coding is based on the mapping colors in fig. A+B, therefor I only changed the line thickness so that the spectrum is well visible in yellow.

**6) Line 84-88: Delete, as this is repeating what noted earlier. Add the last sentence to the end of the previous paragraph. "The spectral libraries...." after the url address.**

This does not seem a repetition for us, as it just least through the chapters of the manuscript and aims to explain the structure. The previous paragraph explains which is provided in the scope of this datapaper + data publications.

**7) Table 1 is not really necessary-you can summarize that info in the text.**

The table is supposed to give a quick overview of the included supplements and is easier to read.

**8) Table 2-4  Suggest combining Tables 2, 3 and 4 into 1 as many parameters are common. So you can have more columns relative to each of the different mineral groups.**

As these tables relate to measurements of three different sets of samples and three different data description reports, these tables should stay separated. The measurement parameters are also provided separately in Excel TM files in the data.

**9) Table 5: The header "Concentration level determination" does not relate to what you have placed in the rows. What are they? All you do is re-reference the data sets. Likely this Table is not critical to the paper.**

We changed tis geochemical analysis. Again, this table should only give an easy to read overview of the different geochemical analyses used for the different sample types

**10) Line 227-233: This is repeated in the data set paper, why include this detail here. If this is the preferred place, then shorten that in the data set paper. Principles of XRF analysis are not really needed, can refer to a paper on this subject**

As we explained all instruments used, e.g HySpex, we aimed to explain every analysis method shortly in order to save the reader from cross referencing too many papers. The data set papers are only "technical reports" on the gfz data services website. The data should be understandable by just accessing this website without knowledge of the datapaper.

**11) Table 6: What about adding in the probable interferences for analysing the LREE with a hand- held XRF instrument. This provides very little information to the reader.**

The XRF analysis is based on Bösche, 2015 dissertation with more informations on the XRF measurements. We've now added the citation.

**12) List background count times on EPMA, especially for the trace elements.**

The REE EPMA method is based on Lorenz et al. 2019 and was developed by her at the university as part of her Ph.D. we are not able to forward the method before her Ph.D. is finished.

**13) Line 254: Measuring F in fluorapatite is problematic using EPMA. At a minimum, the Area-Peak Factors method should be used otherwise result is usually spuriously-high values.**

see Lorenz et al., 2019 "Fluorine, as one of the least-stable components in apatite, was measured first during the analytical scheme. To reduce halogen migration, counting times for F were reduced to 6 s on peaks and 3 s on backgrounds. The EMPA data were reduced with the PRZ-XXP correction routine.

**14) Line 268-274: delete, and any additional information incorporate with the EPMA section.**

EMPA explanation is done for the two different sample groups with different parameters and correction routines (copper and REE).

**15) Line 290: why was one sample analysed using "aquatic" What is so different about it-this is worth explaining. The code info is better placed in the data sheets information.**

We deleted the tables from the data paper and instead added them in the data description of the technical report. The info for the aquatic analysis is also added in there. "Sample 1a belongs to a group of samples where only the weathered crust was extracted for analysis. Only sample 1a is provided here. The aquatic analysis has a very low detection limit and was chosen due to the expected depleted element content."

**16) Table 8 and 9: Is this really important? You have not noted any samples numbers in the text so there is information for the reader in table 8. This should be in an Appendix if deemed critical. Describe the analyses and how they are done –that is more important.**

The tables were deleted from the paper.

**17) References Turner et al, 2014a is out of order, Turner 2015 should be first (single author). Leave a space between Tong et al and the turner reference.**

this is based on the ESSD template and resulting reference list from mendeley desktop

**18) Line 310-314: Only certificate info provided as validation for the REO powders For REE minerals, there is no table comparing the results from the XRF vs. EPMA and indicating if the**

**results are actually comparable. For the geochem analyses by BVM there is no information on how good the data is, no comparison of duplicates, no CRM's, unless this is also held in the University site.**

REO powders are only characterized by the certificates which can be found in the supplements. XRF and EPMA will not be compared as the scope of this paper is only to provide data and not to interpret/compare them. The standards measured by BVM are provided in the Excel TM files, apparently the standard measurements reached the expected results therefor no information on the quality of the analysis was given by BVM/ is necessary.

| Sample | decimal latiti | decimal long | Method | WGHT | LF300 | LF300 | LF300 | LF300 | LF300 | LF300 | LF300 | LF300 | LF300 |
|---|---|---|---|---|---|---|---|---|---|---|---|---|---|
| | | | Analyte | Wgt | SiO2 | Al2O3 | Fe2O3 | MgO | CaO | Na2O | K2O | TiO2 | P2O5 |
| | | | Unit | KG | % | % | % | % | % | % | % | % | % |
| | | | MDL | 0.01 | 0.01 | 0.01 | 0.04 | 0.01 | 0.01 | 0.01 | 0.01 | 0.01 | 0.01 |
| | | | Type | | | | | | | | | | |
| AP/1-A - 1f | 35,077 | 32,84275 | Soil | 0.07 | 45.44 | 15.86 | 12.79 | 5.11 | 7.65 | 2.37 | 1.01 | 1.00 | 0.07 |
| AP/1-A - 2a | 35,076867 | 32,84275 | Soil | 0.11 | 40.72 | 8.00 | 31.22 | 2.67 | 0.75 | 0.31 | 0.67 | 0.38 | 0.20 |
| AP/1-A - 3a | 35,076983 | 32,843083 | Soil | 0.11 | 44.66 | 14.20 | 15.25 | 4.87 | 3.74 | 1.15 | 0.86 | 0.74 | 0.05 |
| AP/1-A - 3b | 35,077 | 32,84305 | Soil | 0.08 | 49.15 | 11.83 | 13.91 | 5.40 | 3.59 | 0.85 | 0.51 | 0.58 | 0.05 |
| AP/1-A - 4a | 35,076967 | 32,843067 | Soil | 0.03 | 33.97 | 10.29 | 10.62 | 9.96 | 8.00 | 0.22 | 0.02 | 0.24 | 0.01 |
| AP/1-A - 4b | 35,077 | 32,843033 | Soil | 0.06 | 42.68 | 16.01 | 13.35 | 8.08 | 2.80 | 0.83 | 0.42 | 0.50 | 0.02 |
| AP/1-A - 6a | 35,076967 | 32,8431 | Soil | 0.09 | 44.32 | 12.76 | 19.85 | 4.44 | 0.99 | 0.65 | 1.77 | 0.42 | 0.03 |
| AP/1-A - 6b | 35,07695 | 32,8432 | Soil | 0.13 | 45.93 | 14.81 | 14.24 | 5.49 | 5.43 | 1.63 | 0.85 | 0.79 | 0.07 |
| AP/1-A - 6c | not available | not available | Soil | 0.11 | 43.27 | 8.94 | 25.45 | 4.54 | 0.79 | 0.31 | 0.48 | 0.27 | 0.02 |
| AP/1-A - 6d | not available | not available | Soil | 0.11 | 39.93 | 12.47 | 24.96 | 4.30 | 1.19 | 0.53 | 1.24 | 0.41 | 0.06 |
| AP/1-A - 13b | 35,076117 | 32,8434 | Soil | 0.12 | 57.20 | 13.23 | 14.08 | 3.77 | 0.71 | 0.35 | 0.08 | 0.82 | 0.03 |
| Pulp Duplicates | | | | | | | | | | | | | |
| AP/1-A - 6d | not available | not available | Soil | 0.11 | 39.93 | 12.47 | 24.96 | 4.30 | 1.19 | 0.53 | 1.24 | 0.41 | 0.06 |
| AP/1-A - 6d | not available | not available | REP | | | | | | | | | | |
| AP/1-A - 4a | 35,076967 | 32,843067 | Soil | 0.03 | 33.97 | 10.29 | 10.62 | 9.96 | 8.00 | 0.22 | 0.02 | 0.24 | 0.01 |
| AP/1-A - 4a | 35,076967 | 32,843067 | REP | | 34.28 | 10.20 | 10.53 | 9.92 | 7.93 | 0.22 | 0.02 | 0.24 | <0.01 |
| Preparation Duplicates | | | | | | | | | | | | | |
| AP/1-A - 4b | 35,076967 | 32,843067 | Soil | 0.06 | 42.68 | 16.01 | 13.35 | 8.08 | 2.80 | 0.83 | 0.42 | 0.50 | 0.02 |
| AP/1-A - 4b | 35,076967 | 32,843067 | DUP | | 42.54 | 16.09 | 13.35 | 8.12 | 2.81 | 0.83 | 0.43 | 0.50 | 0.02 |
| Reference Materials | | | | | | | | | | | | | |
| STD GS311-1 | | | STD | | | | | | | | | | |
| STD GS910-4 | | | STD | | | | | | | | | | |
| STD SO-19 | | | STD | | 60.47 | 13.98 | 7.38 | 2.94 | 5.95 | 4.07 | 1.31 | 0.70 | 0.31 |
| STD SO-19 | | | STD | | 60.60 | 13.91 | 7.47 | 2.90 | 5.92 | 4.02 | 1.30 | 0.70 | 0.32 |
| STD SO-19 | | | STD | | 60.23 | 14.03 | 7.60 | 2.95 | 5.99 | 3.98 | 1.29 | 0.70 | 0.31 |
| STD SO-19 | | | STD | | 60.51 | 13.95 | 7.47 | 2.92 | 5.96 | 4.02 | 1.29 | 0.70 | 0.30 |
| BLK | | | BLK | | | | | | | | | | |
| BLK | | | BLK | | 0.03 | <0.01 | <0.04 | <0.01 | <0.01 | <0.01 | <0.01 | <0.01 | <0.01 |
| BLK | | | BLK | | 0.01 | <0.01 | <0.04 | <0.01 | <0.01 | <0.01 | <0.01 | <0.01 | <0.01 |
| Prep Wash | | | | | | | | | | | | | |
| QUARTZ_KRA | | | Prep Blank | | 98.24 | 0.52 | 0.76 | 0.03 | <0.01 | <0.01 | 0.12 | 0.09 | <0.01 |

**19) Line 337: provide information on how to obtain easy access to this geochemical data.**

This based on the firewall on your computer (?) and not the accessibility of the repository.

Technical Report REE:

**1) What is really needed, is to identify which REE corresponds to which reflection-as in Turner's 2015 thesis (and published papers 2014). This is the test for validation.**
The scope of the data publication is not to identify distinct peaks of different rare earth elements. We only want to provide full spectra for the VNIR and SWIR range in form of a spectral library. The spectrum of each mineral is unique.

**2) I do not see REO analyses in any table, certainly not in Table 5.**
The sample analyses are the purity certificates in table 4.

**3) I could not access geochemical data by XRF nor EPMA.**
Again, this should be in the open access gfz data-service repository

Copper-bearing mineral report

**1) I believe this is the place where that should be held, and maybe the Supplement does not need to be so extensive with images.**
We believe the images should be in both reports. We have a complete data paper describing all samples, therefor it makes sense that these samples are at least shown in the supplements.

**2) For this set of data, I was able to open the EPMA results as well as the SEM images and EDS analyses. I assumed that the EPMA of the 25 analyses match the samples depicted in the SEM file, however that was not the case. It would be very useful in the EPMA files to indicate the sample name using the abbreviations provided (A1, A2, A3, etc.) for cross-reference. Also, as you have several samples of one mineral, averaging their mineral chemistry and calculating an actual mineral formula would be most useful. This can be also calculated for the individual samples.**
We did not average the three measurements of the EPMA as the data publication is supposed to provide raw, unchanged data. We did however provide 3 EPMA measurements of each sample seen the SEM images if the user wants to average the analysis he can do so. In the SEM PDF, the averaged EPMA is provided in comparison with the SEM.

**3) What is the XRD trace of this mineral?**
XRD trace is not provided for any of the samples in the datapaper. It is not a new mineral but based on our measurements we do not feel confident to notate it. Instead we provide the geochemistry and spectrum.

**4) No issues with the spectral data; can open the data files. What validation has been done between the spectral signature of a samples and its geochemistry?**
None, we only provide the data

**5)Why not summarize the chemistry in a table and indicate calculated formula for each mineral? What are the ore minerals?**

Because the EPMA analysis was done in two programs for the oxides and sulphides + native copper separately. Therefor the analysis will be given separately. The ore minerals are the sulphides and native copper.

Apliki samples

**1) Table 2:** we added the missing information

**2) Table 3 and Table 4 were also in the main paper-they are not really important here; provide a web address. I think a description of the method is more valuable to the reader. Also, need to provide accuracy and precision information, there is no mention of duplicate or CRMs being analysed. Geochem data could not be accessed.**
We cut the tables from the main paper and provided them here. As each analysis group has different steps of analysis but some of them are similar it makes sense to us to provide the analysis methods in a table format. The data has to be accessible.

**3) What validation has been done between the spectral signature of a samples and its geochemistry?** None, in this paper we only describe the data that is accessible, no interpretation whatsoever is provided either here or in the data datapaper.

**Response to reviewer's comments (D. Turner)**

We respond here to the comments that needed a more detailed answer other than addressing them in the publication itself.

**I) Datapaper + Supplements – general comments:**

**1) While it is clear that there are linkages between the datasets, there is not linkage between all the topics. For example, the REE-related spectra match well with the Cu-related spectra, but not the Apliki site data. It is this reviewer's opinion that two ESSD papers could be written, perhaps best to match the REE and Cu mineral-focused spectra together in a single paper, and the Apliki site data by itself. That way, the target information is more consistent within the described datasets. In the present format, the four suites of data are all different in the physical nature of the samples and the (geo)chemical and mineral characterization.**

We are able to comprehend the reviewers argumentation and realize the difference between the here presented data sets. As we are planning to publish more spectral data of different origin and sample characteristica, the data paper is supposed to be an comprehensive overview of the spectral data acquisition in the GFZ laboratory. The datasets themselve do not have to have the same geochemical or mineral characterization, as their physical state and geochemicstry is separately described in the data reports themselves. We therefore argue that the datapaper is presented as one conclusive umbrella publication that can be referred to when accessing the spectral data and geochemical data itself.

**2) The manuscript starts with some fairly sweeping comments in the introduction that could be toned down in the context of this contribution.**

You're completely right! We acknowdledge the enthusiastic tone relating to our contribution to the accredited USGS libraries and toned it down. The rest of the introduction is a review of the associated methods and to the best of our knowledge in a neutral tone. When re-reading the manuscript we tried to re-structure section parts that detracted the reader.

**3) In order for the broader community to use any spectral library, documentation needs to be clear and there should be no ambiguity regarding data source or characterization methods.**

Your comments regarding the document were taken seriously and we cleared up the document based on your indications regarding the state of purity of the spectra, the mineral structure and the different geochemical analysis types. This should help the reader to clearly catch on the scope of this work. Again, the aim was to describe the

hyperspectral data acquisition in the laboratory that is consistent for all sample types. The geochemical analysis differences have been explained in detail (in the main paper and the data reports) and is dependent on the target material.

**4) In this sense, I also recommend splitting the paper into two discrete contributions so as to reduce the potential confusion over what methods were used on which samples.**

It is our aim to show the hyperspectral data acquisition and explain in detail how the hyperspectral laboratory work is conducted. The geochemical data for validation is not supposed to be of the same method of non-ambiguous.

**5) Furthermore, the samples that are rocks and not minerals need to be identified as such. For example, the monazite and synchysite spectra are not what I would have expected if these were mineral specimens. Since one of the principle uses of these spectra are as inputs to understand unknown spectra in other datasets, it is important to state their true nature.**

We cleared this in the data report as well as the ESSD paper. The REE-bearing minerals/ minerals within a rock matrix. All of these samples are supplied with XRF data but not SEM. As these samples have been part of a number of diverse publications within the Ph.D. project of N.K. Bösche, and have been published and peer-reviewed, we did not re-analysis the samples. As the results based on these data (Bösche, 2015) and successful publications we wanted to share them with a broader community.

**II) Datapaper Comments**

**Line 70. REE oxides are not minerals, though they are likely crystalline. Do you know the structure of these REE Oxides?**

You are correct, the REE-oxides are synthetic powders not in crystalline structure.

**Line 104. How did you validate the mineral species? (is related to the REE-bearing minerals)**

The minerals were not validated, the notation based on the supplier was assumed valid. The supplier (http://www.seltene-mineralien.de) offers analytical services with a modern REM-EDX technology and therefor we assume his specimen are analysed and the mineral species is validated before the sale.

**Line 108. How did you validate the mineral species?  (is related to the Copper-bearing minerals)**

Mineral identification took place using microprobe analyses.

**Table 1. Why two suites for Copper bearing minerals? State why.**

Both supplemental entries are for one suite of samples. S4 is a table stating the sample origin, whereas S5 lists the sample Ids and photos.

**Line 345. 120 seconds isn't really short for a HHXRF**

The long duration of 120 seconds were chosen with the purpose to reduce the noise of the measurement.

**Line 121. "The sample preparation varied by sample type and depends on the material and the information of interest." This is a bit problematic. It is not that any one of the approaches isn't valid, but the presentation of the data as a collection should strive to have internally consistent methods.**

The hyperspectral data acquisition described in this document is the internally consistent method that is presented here. We are planning to publish more hyperspectral libraries in the future. All of these hyperspectral libraries will be compiled under the conditions as described here. The geochemically analysis is different not only based on the sample type but the level of detail required for the projects where our spectral libraries stem from. The geochemical analysis is presented only as a source of validation not as a data publication itself. Unfortunately our funding does not allow for the compilation of spectral libraries with consistent geochemical validation as the likes of  the USGS Spectral Library Version 7 (Kokaly et al., 2017).

**Section 5.2 Why do you have two distinct descriptions for the JEOL unit at Potsdam?**

The University of Potsdam used two JEOL units, one for SEM and one for EMPA analyses. The EMPA analyses is described twice, one each for the sample type analyzed, as different measurement parameters were used.

**Line 272. Which standards? Same as above? If so, many of the Smithsonian materials are synthetic.**

SEM measurements are calibrated with pure copper and the EMPA measurements are calibrated with the standards mentioned in the text from the Smithsonian Institution and Astimex.

**III) Apliki technical report:**

**To confirm above points, the "aquatic" sample above is not an aquatic sample. Instead, it was analyzed by aqua regia digestion, as described in the BVM analytical notes. Why was this one sample analyzed with a different method? The manuscripts need to have this item addressed.**

> The internal BVMs sample preparation-/ analysis type groups by the required analysis method. These "analysis types" are namely "aquatic", "rock" and "soil". The measurement/ analysis type is chosen by the service provider "BVM Institute" on which we do not have any influence.

**It is unfortunate that your overlimit Cu samples were not re-analyzed for total Cu content, since this is a key focus for the Cu mine related samples.**

> Yes, the BVM analysis limits were chosen in order to resolve sample copper content with lower limits. We honestly did not expect to have three samples with Cu content > 10.000 ppm. Unfortunately, the analysis could not be repeated.

**Figure 2. Maybe you could split this out into two plots, rocks and soils, so that the reader can maybe make a small assessment of the data without downloading and plotting the spectra?**

> We've given the plot a larger space in document. Additionally the spectra are presented and shown in more detail in the Ph.D. thesis of Koerting (2020, in preparation).

**IV) Copper-bearing technical report:**

**Sample C3 has a typo in its name, where a "%" is used instead of "5"**

Unfortunately, we cannot find the typo, maybe it was fixed during our reworking of the document from review 01

**There are some samples that are not monomineralic, and this is important to state in the documentation. For example, "Malachite 1" has a whack of other minerals in the SEM image, so right away we know that the "Malachite" spectrum here is actually at least Malachite+pyroxene+quartz. Same for Azurite 2 and Chalcopyrite 3.**

You are absolutely right, the sample names were used from the names of the collection where the samples were retrieved from. We are adding this information to the technical report. All identified mineral phases of each sample are presented in the file "copper_bearing_minerals_chemistry.pdf" in the published data. The samples in question are: A1, A3, L1, P3, B1, M2 & C3

**V) REE technical report:**

**You should include the anticipated crystal structure of these oxides. The work by White (1965?... off the top of my head) showed the importance of this variable for the resulting spectra. This is especially relevant in the context that Tb is listed as both 3+ and 4+.**

The synthetic REO powders were delivered by the supplier. The REE-bearing minerals and REO powders were  published by (Bösche, 2015; Herrmann, 2019) and are described in detail in the reviewed works.

**Similar to the copper minerals, some of these samples are not monomineralic and instead are rocks, and therefore the spectra need to be identified as such. For example, the images you show in the EMPA data indicate that synchysite has a bunch of phases, as does ilmenite.**

You are absolutely right, the spectra of the REMin where supposed to be acquired only over the identifyable mineral surface in the hyperspectral imagery. The geochemical data indicate though, that some of the sampled area is not monomineralic. The geochemical data is supplied with the samples to show exactly that. We will note this in the geochemical part in the technical report.

**Do you have analyses of the monazite sample?**

Yes, we have the XRF analyses of the sample.

[revised manuscript text omitted]

---

## Editor Decision (ED1)

Requirements are to enhance the usability of the published data collections and the manuscript following the referees' and editorial recommendations.

Please upload the revised version with i) new answers to the referees as in the current version of answers to the reviewers it becomes not clear if and how all referee points have been addressed, e.g. please list what edits and changes you could made in the published data collection

and with ii) the new manuscript version with track changes included.

**In general**

The ESSD manuscript describes a spectral library collection without in this moment visualising the data in form of spectral library plots and without a user-friendly overview on the content per collection.

i) provide more detailed tables on the content of the spectral library collections in the main text, e.g. provide the name of the minerals per collection in overview tables in the ESSD manuscript

ii) provide plots (e.g. stacked or group associated) of the spectral library collections in the main text. You can also add more specific details in the appendix

**data publication**

in the data collection enhance the ascii files. Now the ascii spectral library files are directly exported from the ENVI spectral libraries.

Enhancement of ascii files: you have the possibility to add more codes and explanations: please add – in addition to the ENVI spectral library code name of the reflectance spectra – at least another code that is also clearly the same code in a linked chemical measurement, please add also a column for the general name of the mineral / element.

Specifically for the copper bearing minerals you need a clearly identifying code system so that the geochemistry and reflectance spectra can be linked.

If you intend to keep the Apliki mine spectral library collection in this ESSD manuscript you need to provide a linkage to the other data collections, e.g. you could do this in the form of mineral interpretation

Please consider the Referees comment: The authors should also explain exactly what they mean by geochemical validation-to show the composition of the samples, or to show a relationship between composition and absorption minima?

and make more clear in the text how you apply the term 'validation

**specific requirements**

**Abstract:** please name how many samples are in the specific collections each

**Introduction:** Please change the focus of your introduction: in your manuscript, the introduction leads into the direction of hyperspectral imaging, specifically on future hyperspectral satellite missions such as EnMAP and future imaging spectroscopy applications that could make use of the spectral libraries. While this can be a part of the introduction substantial information is not provided yet.

Please introduce in the introduction the REE- and copper bearing target minerals / rock assemblages that are your published data collection in which ore deposits they occur and for what these elements are used for Explicitly show which spectral libraries on REE and / or copper bearing target minerals are available, e.g. USGS, NASA-Jet Propulsion Laboratory Library, Canada Geological Survey published dataset (with citations)

The description of the future ENMAP satellite acquisition principle is not relevant for this ESSD publication, please delete:
' The image scenes are acquired by a moving line scanner mounted on the satellite, which records the spatial dimension (x- and y-dimension) line by line, as well as the wavelength dimension (z-dimension). Each pixel therefore represents the full spectral range of the sensor. The sensor's movement along a rotation or a movement axis provides spatially continuous imaging spectroscopy data.'

Better to show that spectral reflectance measurements and characteristics are clearly based on physics by citing some literature on mineral spectroscopy and also the reason of absorption in the VIS (electronic transitions) and the SWIR (molecular vibrations) instead of to descriptive sentences e.g. 'Variations along the spectral domain of the data are visible as concave indentions, often referred to as "absorption bands". – They are absorption bands

L36 in this study -> change into this data collection

**Methods:** please combine chapters 3, 4 and 5 to one chapter method with subchapters – may be ad more tables

add a chapter '**results**'

Here you can provide plots (e.g. stacked or group associated) of the spectral library collections

**Discussion –**

You provide discussion chapters on accuracies

Here is also the opportunity on a short chapter for an overview what your data collection provides to the user of the data collections, where you can better bring the Apliki spectral library into the context of this manuscript.

Please note: there is nowhere a requirement about spectral analyses on spectral characteristics.

Throughout the text: reflectance spectra (instead of spectra)

---

## Author Response (AR4)

**1. Response to editor's comments**

**1) Reply to comments in Feb. 2021**

**Comment 1: Please modify or discard Figure 13 Figure 13 shows examples of reflectance spectra, however, it remains unclear to the reader what is displayed: the naming of the spectra are numbers that do not appear as codes in the table, the Y-axis does not show the scaling of the unit of reflectance (that would be here 0 -1 * scaling factor), you could either add in figure 13 the labels that you also present in the tables or discard figure 13.**

Answer: Figure 13 has been modified; the full sample names have been added in the legend (e.g. "Apl1_A_13b [5x5AVG]" instead of "13b")
The data scaling has been added in the caption (between 0-10000).

**Comment 2: Please add information on the scaling factor that you use for your reflectance data as - a sentence in the manuscript, e.g. would fit well in 3.3 Hyperspectral Data Processing - a sentence in your data publication abstracts at GFZ data services reflectance is expressed either as 0-100 % or 0 -1 unitless (as the theoretical ratio of outgoing energy / radiance versus incoming energy / radiance) in the case of your published data sets, reflectance is expressed unitless with a scaling factor that needs to be described in the manuscript text and in the data publication abstracts.**

Answer: The scaling from 0 – 10.000 is based on the HySpex sensor calibration and the Reflectance Retrieval Routine described in Rogass et al., 2017. We added the information regarding this unitless scaling in Chapter 3.3 of the manuscript as well as in the caption of Figure 13, where the scaling of the spectra appears. As the other figures showing the spectral libraries are stacked, here the scaling is not shown.

The information regarding the scaling were added to the data reports hosted on the GFZ Data Service website and are already online. This information is added in the Abstract as well as the "Hyperspectral Measurement" section.

**2) Reply to comments in Nov. 2020**

**General comment by the authors: As the repetition of information in the supplements and the technical report seems to cause confusion, we have decided to delete the supplements and only refer to the technical reports that present all of the needed detailed information regarding the samples.**

Question: Requirements are to enhance the usability of the published data collections and the manuscript following the referees' and editorial recommendations.

Question: Please upload the revised version with i) new answers to the referees as in the current version of answers to the reviewers it becomes not clear if and how all referee points have been addressed, e.g. please list what edits and changes you could made in the published data collection
*Answer: the revised answers will be uploaded as "_comments_v2"*

Question: and with ii) the new manuscript version with track changes included.
*Answer: We will do that and upload them separately*

**I) In general**

Q: The ESSD manuscript describes a spectral library collection without in this moment visualising the data in form of spectral library plots and without a user-friendly overview on the content per collection.
i) provide more detailed tables on the content of the spectral library collections in the main text, e.g. provide the name of the minerals per collection in overview tables in the ESSD manuscript

ii) provide plots (e.g. stacked or group associated) of the spectral library collections in the main text. You can also add more specific details in the appendix
*A: Added these info in the manuscript from line 510 to line 548.*
*We added a chapter 4 "results", presenting the spectral libraries, the spectral entries and a plot for each spectral library. All of this information can be found in more detail in the technical reports provided on the data platform and we're again citing these technical reports.*

**II) Data publication**

Q: in the data collection enhance the ascii files. Now the ascii spectral library files are directly exported from the ENVI spectral libraries.
Enhancement of ascii files: you have the possibility to add more codes and explanations: please add – in addition to the ENVI spectral library code name of the reflectance spectra – at least another code that is also clearly the same code in a linked chemical measurement, please add also a column for the general name of the mineral / element.
*A: The ascii files are directly being exported from the spectral libraries in ENVI. The ASCII files and samples are now better described in the technical reports of the data. In addition, the header in the ASCII files define the columns in the data file. The tables in the technical reports list all these information for each sample and are provided on the DOI landing page of the data publication.*

Q: Specifically for the copper bearing minerals you need a clearly identifying code system so that the geochemistry and reflectance spectra can be linked.
*A: This has been done in the new technical reports and spectral libraries that will be uploaded as version 2 of the data and technical reports. Here, each technical report lists the samples of the spectral library in a table. We have added a column with the spectrum name for each sample for the copper and REO and REMin samples, now all samples (including the Apliki samples) are clearly named and the spectra names clearly identify the samples. In addition, we have included sample overview tables of all data publications in the manuscript section 4.*

Q: If you intend to keep the Apliki mine spectral library collection in this ESSD manuscript you need to provide a linkage to the other data collections, e.g. you could do this in the form of mineral interpretation
*A: Lines 583 – 587,  we've added that the sample collections themselves are independent from each other in line 219 – 262 (the line numeration is off, somehow this happens in the template?). Our manuscript aims to explain the method of spectral data collection by the HySpex system and present different datasets for a wider community based on the HySpex sensor characteristics. However, to make the description of the Apliki samples more comparable with the copper and REE samples, we have added a general description of the sulphide deposits of the Apliki Mine to the Technical Report and an additional column describing the mineralogy based on qualitative XRD analysis in Table 1 of the Technical Report.*

Q: Please consider the Referees comment: The authors should also explain exactly what they mean by geochemical validation-to show the composition of the samples, or to show a relationship between composition and absorption minima?
and make more clear in the text how you apply the term 'validation
*A: we've added this in line 214-218: "The samples are presented as reflectance spectral libraries and their geochemical composition. Sample nominations are based on the geological collection of origin or sample abbreviations from the field sampling. The sample nomination is not an interpretation of the presented geochemical data. The datasets are independent of each other, the reflectance spectra can be seen as a spectral expression of the existing geochemical data. Neither the geochemistry nor the reflectance spectra are interpreted or correlated to each other.*

**III) specific requirements:**

**Q: Abstract:** please name how many samples are in the specific collections each
*A: added in lines 23 - 29, "The spectral libraries and corresponding geochemistry are published via GFZ Data Services with the following DOIs: http://doi.org/10.5880/GFZ.1.4.2019.004 (13 REE-bearing minerals and 16 oxide powders, Koerting et al. (2019a)), http://doi.org/10.5880/GFZ.1.4.2019.003 (20 Copper-bearing minerals,*

*Koellner et al. (2019)), and [http://doi.org/10.5880/GFZ.1.4.2019.005](http://doi.org/10.5880/GFZ.1.4.2019.005) (37 Copper-bearing surface material samples from the Apliki copper-gold-pyrite mine in Cyprus, Koerting et al. (2019b)). All spectral libraries are united and comparable by the internally consistent method of hyperspectral data acquisition in the laboratory."*

**Q: Introduction:** Please change the focus of your introduction: in your manuscript, the introduction leads into the direction of hyperspectral imaging, specifically on future hyperspectral satellite missions such as EnMAP and future imaging spectroscopy applications that could make use of the spectral libraries. While this can be a part of the introduction substantial information is not provided yet.
*A: We have changed the order of the different sections within the introduction. It should now be clear why hyperspectral satellite missions are needed for the geology applications that we refer to. Here, specifically from line 145 - 154.*

Q: Please introduce in the introduction the REE- and copper bearing target minerals / rock assemblages that are your published data collection in which ore deposits they occur and for what these elements are used for  Explicitly show which spectral libraries on REE and / or copper bearing target minerals are available, e.g. USGS, NASA-Jet Propulsion Laboratory Library, Canada Geological Survey published dataset (with citations)
*A: In our opinion a full intro into the different spectral libraries goes too much into detail, similar to a review of the existing libraries. We've added why our spectra and samples can contribute to commonly known libraries and how we aim to make them available for other hyspex users. This can be found in line 156 - 162.*

Q: The description of the future ENMAP satellite acquisition principle is not relevant for this ESSD publication, please delete:
' The image scenes are acquired by a moving line scanner mounted on the satellite, which records the spatial dimension (x- and y-dimension) line by line, as well as the wavelength dimension (z-dimension). Each pixel therefore represents the full spectral range of the sensor. The sensor's movement along a rotation or a movement axis provides spatially continuous imaging spectroscopy data.'
*A: this has been deleted, as suggested.*

Q: Better to show that spectral reflectance measurements and characteristics are clearly based on physics by citing some literature on mineral spectroscopy and also the reason of absorption in the VIS (electronic transitions) and the SWIR (molecular vibrations) instead of to descriptive sentences e.g. 'Variations along the spectral domain of the data are visible as concave indentions, often referred to as "absorption bands". – They are absorption bands
*A: We've added the information and citations about mineral spectroscopy from line 43 - 89.*

Q: L36 in this study -> change into this data collection
*A: This has been changed*

**Q: Methods:** please combine chapters 3, 4 and 5 to one chapter method with subchapters – may be add more tables

*A: Has been changed and adjusted*

Q: add a chapter '**results**'.  Here you can provide plots (e.g. stacked or group associated) of the spectral library collections
*A: We added a chapter 4 "results", listing the samples, relevant information, spectra names and a plot for each spectral library. All of this information can be found in more detail in the technical reports provided on the data platform and we're again citing these technical reports.*

**Q: Discussion –**
You provide discussion chapters on accuracies
Here is also the opportunity on a short chapter for an overview what your data collection provides to the user of the data collections, where you can better bring the Apliki spectral library into the context of this manuscript.
*A: Added briefly in line 636 - 640*

Q: Please note: there is nowhere a requirement about spectral analyses on spectral characteristics.
*A: It's not clear to us what you mean here. Do you mean that we do not discuss the spectral characteristics or correlate spectral features to the minerals from the collections or their geochemistry? We do not aim to discuss the spectra, we simply aim to provide the spectra of each material and the geochemistry of the same material without interpreting if e.g. the mineral denomination from a certain collection is correct or if the spectrum of the sample corresponds correctly to the geochemical data.*

Q: Throughout the text: reflectance spectra (instead of spectra)
*A: Has been cleared up/ "reflectance" was added*

**2.  Authors responses to Jeanne Percival - Reviewer Comments:**

(Comments in bold, answers in regular script.)

**I) Datapaper + Supplements**

**1) The authors should also explain exactly what they mean by geochemical validation-to show the composition of the samples, or to show a relationship between composition and absorption minima?**

In order to avoid confusion with the term validation, we now refer to the geochemical analyses as a possibility for the user to check/interpret the spectra. The hyperspectral spectra and corresponding geochemistry are independent of each other, the reflectance spectra can simply be seen as a spectral expression of the existing geochemical data. Neither the geochemistry nor the reflectance spectra are interpreted or correlated to each other.

**2) These data sets should be available as the spectral library was made available, or somewhere there should be a note as to how to access the files as a guest.**

The data is openly available on the GFZ data service website and licenced with a CC BY 4.0 Licence. There might have been an issue with the firewall of the user's computer or with server maintenance? We had a system update in spring 2020 at GFZ Data Services.

**3) "elemental" from a hyperspectral viewpoint, I would see Fe as the main element observed (i.e., in gossans), as REE-bearing minerals tend to be in trace amounts, very small, hence seeing the NIR signature may be masked by other minerals such as clays or carbonates etc.**

This is just meant as a basic introduction for hyperspectral data uses and not tailored to outcrops or weathered surfaces. The mapping meant here is not only for rock outcrop scan or satellite imagery but also in laboratory scans and in these scans, we do have REE mineral specimen that need to be mapped. Here, the spectrum depends on the element content and is not covered by clay etc. as if found in the field. You are right, in field imagery, the major element mapped is Fe but in laboratory scans, other elements like Nd embedded in monazite can be mapped.

**4) Line 46: should references be in chronological order from oldest to youngest? + Turner et al 2014a is out of order in the ref list.**

We've changed this in the references

**5) Fig. 1: This is very good and provides a good image of what your paper is about. Suggest that the spectral pattern for chalcopyrite be a darker colour, cannot see the yellow; also the malachite should be a continuous line like the other three.**

We changed the malachite to a continuous line. The color-coding itself is based on the mapping colors in fig. A+B. We changed the line thickness so that the spectrum is well visible in yellow but corresponds to the colors from A and B.

**6) Line 84-88: Delete, as this is repeating what noted earlier. Add the last sentence to the end of the previous paragraph. "The spectral libraries...." after the url address.**

We hoped that this part helps to lead the reader through the chapters of the manuscript and aims to explain the structure. The previous paragraph has been rewritten, so there is not repetition anymore (see lines 220 - 226)

**7) Table 1 is not really necessary-you can summarize that info in the text.**

We decided to delete the supplements, as the information is repetitive to the technical reports of the dataset and seem to have been confusing. Therefore, Table 1 was also deleted and any reference to the supplements.

**8) Table 2-4 Suggest combining Tables 2, 3 and 4 into 1 as many parameters are common. So you can have more columns relative to each of the different mineral groups.**

We changed that. Now we show the common parameters in Table 1 and the different dataset parameters in Table 2-3 (page 10, starting at line 390).

**9) Table 5: The header "Concentration level determination" does not relate to what you have placed in the rows. What are they? All you do is re-reference the data sets. Likely this Table is not critical to the paper.**

We changed this to "geochemical analysis". This table should give an easy to read overview of the different geochemical analyses used for the different sample types

**10) Line 227-233: This is repeated in the data set paper, why include this detail here. If this is the preferred place, then shorten that in the data set paper. Principles of XRF analysis are not really needed, can refer to a paper on this subject**

As we explained all instruments used, e.g HySpex, we aimed to explain every analysis method shortly in order to save the reader from cross referencing too many papers. We think that  the data should be understandable by just accessing the Gfz Data Service website, where the technical reports are provided, without knowledge of the data paper.

**11) Table 6: What about adding in the probable interferences for analysing the LREE with a hand- held XRF instrument. This provides very little information to the reader.**

The XRF analysis is based on Bösche (2015) dissertation with more information on the XRF measurements. We've now added the citation.

**12) List background count times on EPMA, especially for the trace elements.**

The REE EPMA method is based on Lorenz et al. (2019) and was developed by her at the university as part of her PhD Thesis. Therefore, we are not able to forward the method before her Ph.D. is finished.

**13) Line 254: Measuring F in fluorapatite is problematic using EPMA. At a minimum, the Area-Peak Factors method should be used otherwise result is usually spuriously-high values.**

Lorenz et al., 2019  say "Fluorine, as one of the least-stable components in apatite, wasmeasured first during the analytical scheme. To reduce halogen migration, counting times for F were reduced to 6 s on peaks and 3 s on backgrounds. The EPMA data were reduced with the PRZ-XXP correction routine".

**14) Line 268-274: delete, and any additional information incorporate with the EPMA section.**

We added both EPMA explanations together now, it is done for the two different sample groups with different parameters and correction routines (copper and REE) (lines 466 - 529, section 3.4.2)

**15) Line 290: why was one sample analysed using "aquatic" What is so different about it-this is worth explaining. The code info is better placed in the data sheets information.**

We deleted the tables from the data paper and instead added them in the data description of the technical report. The info for the aquatic analysis is also added in there. "Sample 1a belongs to a group of samples where only the weathered crust was extracted for analysis. Only sample 1a is provided here. The aquatic analysis has a very low detection limit and was chosen due to the expected depleted element content."

**16) Table 8 and 9: Is this really important? You have not noted any samples numbers in the text so there is information for the reader in table 8. This should be in an Appendix if deemed critical. Describe the analyses and how they are done –that is more important.**

The tables were deleted from the paper.

**17) References:   Turner et al, 2014a is out of order, Turner 2015 should be first (single author). Leave a space between Tong et al and the turner reference.**

We've redone the references and changed the order.

**18) Line 310-314: Only certificate info provided as validation for the REO powders  For REE minerals, there is no table comparing the results from the XRF vs. EPMA and indicating if the results are actually comparable. For the geochem analyses by BVM there is no information on how good the data is, no comparison of duplicates, no CRM's, unless this is also held in the University site.**

REO powders are only characterized by the certificates which can be found in the corresponding technical report As we do not interpret or correlate the different datasets, XRF and EPMA are not compared. The standards measured by BVM are provided in the Excel TM files, apparently the standard measurements reached the expected results. Therefore, no information on the quality of the analysis was given by BVM.

| Sample | decimal latit | decimal long | Method | WGHT | LF300 | LF300 | LF300 | LF300 | LF300 | LF300 | LF300 | LF300 | LF300 |
|---|---|---|---|---|---|---|---|---|---|---|---|---|---|
| | | | Analyte | Wgt | SiO2 | Al2O3 | Fe2O3 | MgO | CaO | Na2O | K2O | TiO2 | P2O5 |
| | | | Unit | KG | % | % | % | % | % | % | % | % | % |
| | | | MDL | 0.01 | 0.01 | 0.01 | 0.04 | 0.01 | 0.01 | 0.01 | 0.01 | 0.01 | 0.01 |
| | | | Type | | | | | | | | | | |
| AP/1-A - 1f | 35,077 | 32,84275 | Soil | 0.07 | 45.44 | 15.86 | 12.79 | 5.11 | 7.65 | 2.37 | 1.01 | 1.00 | 0.07 |
| AP/1-A - 2a | 35,076867 | 32,84275 | Soil | 0.11 | 40.72 | 8.00 | 31.22 | 2.67 | 0.75 | 0.31 | 0.67 | 0.38 | 0.20 |
| AP/1-A - 3a | 35,076983 | 32,843083 | Soil | 0.11 | 44.66 | 14.20 | 15.25 | 4.87 | 3.74 | 1.15 | 0.86 | 0.74 | 0.05 |
| AP/1-A - 3b | 35,077 | 32,84305 | Soil | 0.08 | 49.15 | 11.83 | 13.91 | 5.40 | 3.59 | 0.85 | 0.51 | 0.58 | 0.05 |
| AP/1-A - 4a | 35,076967 | 32,843067 | Soil | 0.03 | 33.97 | 10.29 | 10.62 | 9.96 | 8.00 | 0.22 | 0.02 | 0.24 | 0.01 |
| AP/1-A - 4b | 35,077 | 32,843033 | Soil | 0.06 | 42.68 | 16.01 | 13.35 | 8.08 | 2.80 | 0.83 | 0.42 | 0.50 | 0.02 |
| AP/1-A - 6a | 35,076967 | 32,8431 | Soil | 0.09 | 44.32 | 12.76 | 19.85 | 4.44 | 0.99 | 0.65 | 1.77 | 0.42 | 0.03 |
| AP/1-A - 6b | 35,07695 | 32,8432 | Soil | 0.13 | 45.93 | 14.81 | 14.24 | 5.49 | 5.43 | 1.63 | 0.85 | 0.79 | 0.07 |
| AP/1-A - 6c | not available | not available | Soil | 0.11 | 43.27 | 8.94 | 25.45 | 4.54 | 0.79 | 0.31 | 0.48 | 0.27 | 0.02 |
| AP/1-A - 6d | not available | not available | Soil | 0.11 | 39.93 | 12.47 | 24.96 | 4.30 | 1.19 | 0.53 | 1.24 | 0.41 | 0.06 |
| AP/1-A - 13b | 35,076117 | 32,8434 | Soil | 0.12 | 57.20 | 13.23 | 14.08 | 3.77 | 0.71 | 0.35 | 0.08 | 0.82 | 0.03 |
| Pulp Duplicates | | | | | | | | | | | | | |
| AP/1-A - 6d | not available | not available | Soil | 0.11 | 39.93 | 12.47 | 24.96 | 4.30 | 1.19 | 0.53 | 1.24 | 0.41 | 0.06 |
| AP/1-A - 6d | not available | not available | REP | | | | | | | | | | |
| AP/1-A - 4a | 35,076967 | 32,843067 | Soil | 0.03 | 33.97 | 10.29 | 10.62 | 9.96 | 8.00 | 0.22 | 0.02 | 0.24 | 0.01 |
| AP/1-A - 4a | 35,076967 | 32,843067 | REP | | 34.28 | 10.20 | 10.53 | 9.92 | 7.93 | 0.22 | 0.02 | 0.24 | <0.01 |
| Preparation Duplicates | | | | | | | | | | | | | |
| AP/1-A - 4b | 35,076967 | 32,843067 | Soil | 0.06 | 42.68 | 16.01 | 13.35 | 8.08 | 2.80 | 0.83 | 0.42 | 0.50 | 0.02 |
| AP/1-A - 4b | 35,076967 | 32,843067 | DUP | | 42.54 | 16.09 | 13.35 | 8.12 | 2.81 | 0.83 | 0.43 | 0.50 | 0.02 |
| Reference Materials | | | | | | | | | | | | | |
| STD GS311-1 | | | STD | | | | | | | | | | |
| STD GS910-4 | | | STD | | | | | | | | | | |
| STD SO-19 | | | STD | | 60.47 | 13.98 | 7.38 | 2.94 | 5.95 | 4.07 | 1.31 | 0.70 | 0.31 |
| STD SO-19 | | | STD | | 60.60 | 13.91 | 7.47 | 2.90 | 5.92 | 4.02 | 1.30 | 0.70 | 0.32 |
| STD SO-19 | | | STD | | 60.23 | 14.03 | 7.60 | 2.95 | 5.99 | 3.98 | 1.29 | 0.70 | 0.31 |
| STD SO-19 | | | STD | | 60.51 | 13.95 | 7.47 | 2.92 | 5.96 | 4.02 | 1.29 | 0.70 | 0.30 |
| BLK | | | BLK | | | | | | | | | | |
| BLK | | | BLK | | 0.03 | <0.01 | <0.04 | <0.01 | <0.01 | <0.01 | <0.01 | <0.01 | <0.01 |
| BLK | | | BLK | | 0.01 | <0.01 | <0.04 | <0.01 | <0.01 | <0.01 | <0.01 | <0.01 | <0.01 |
| Prep Wash | | | | | | | | | | | | | |
| QUARTZ_KRA | | | Prep Blank | | 98.24 | 0.52 | 0.76 | 0.03 | <0.01 | <0.01 | 0.12 | 0.09 | <0.01 |

**19) Line 337: provide information on how to obtain easy access to this geochemical data.**

The data links are provided in the paper, it might have been possible that a server maintenance did not allow you access the data. All data is provided via the DOI link to GFZ Data Services and not another source (e.g. university website or similar).

**II) Technical Report REE:**

**1) What is really needed, is to identify which REE corresponds to which reflection-as in Turner's 2015 thesis (and published papers 2014). This is the test for validation.**
As we do not interpret or correlate the datasets with/to other data we do not provide information to identify distinct peaks of different rare earth elements. We only want to provide full spectra for the VNIR and SWIR range in form of a spectral library. The spectrum of each mineral is unique.

**2) I do not see REO analyses in any table, certainly not in Table 5.**
The sample analyses are the purity certificates in table 5, technical report starting page 9
"*Table 5: **Chemical composition of the REO powders provided by certificates of purity (Bösche, 2015; Herrmann, 2019)**.*"

**3) I could not access geochemical data by XRF nor EPMA.**

All data was and is freely available via the DOI links. GFZ Data Services, however, had a system update probably corresponding with your unsuccessful attempts to access the data. We apologise for this. For the revised version, we have thoroughly restructured the three technical reports and published a new version of the DOI.

**III) Copper-bearing mineral report:**

**1) I believe this is the place where that should be held, and maybe the Supplement does not need to be so extensive with images.**
We have now decided to delete the supplements altogether. Everything is provided in the separate technical reports of the datasets which are closely cross-referenced with the article.

**2) For this set of data, I was able to open the EPMA results as well as the SEM images and EDS analyses. I assumed that the EPMA of the 25 analyses match the samples depicted in the SEM file, however that was not the case. It would be very useful in the EPMA files to indicate the sample name using the abbreviations provided (A1, A2, A3, etc.) for cross-reference. Also, as you have several samples of one mineral, averaging their mineral chemistry and calculating an actual mineral formula would be most useful. This can be also calculated for the individual samples.**

The sample names have all been changed to include the abbreviations (A1_Azurite, A2_Azurite, ..) provided. We did not average the three measurements of the EPMA in the data text-file themselves, as the data publication is supposed to provide raw, unchanged data. A mean however is provided in the table listing all samples (both, in the data paper and the technical report)

**3) What is the XRD trace of this mineral?**
XRD trace is not provided for any of the samples in the article. The copper-bearing samples originate from the university of Potsdam and the BGR and could only be analysed by non-destructive methods. It is not a new mineral but based on our measurements we do not feel confident to notate it. Instead we provide the geochemistry and spectrum.

**4) No issues with the spectral data; can open the data files. What validation has been done between the spectral signature of a sample and its geochemistry?**
None, we only provide the data. The spectrum for each sample is provided, each sample is geochemically described by its geochemical information and in the copper-bearing minerals and the REE samples by the sample denomination provided by the collection holder or the sample re-seller. The reflectance spectra can be seen as a spectral expression of the existing geochemical data. We repeatedly point out now, that the geochemical data should be looked at when denominating the spectra, the sample "names" are only suggestions from the seller or the the originator of the collection (e.g. lines 214 - 218)

**5) Why not summarize the chemistry in a table and indicate calculated formula for each mineral?   What are the ore minerals?**
EPMA and SEM were now combined in the technical report stating the general formula of the mineral that was measured. Pulp duplicates are mentioned in the geochemistry Excel™ file for each BVM analysis method.

**IV) Apliki samples:**

**1) Table 2:** we added the missing information

**2) Table 3 and Table 4 were also in the main paper-they are not really important here; provide a web address. I think a description of the method is more valuable to the reader. Also, need to provide accuracy and precision information, there is no mention of duplicate or CRMs being analysed. Geochem data could not be accessed.**
We seperated the tables from the main paper and provided them here. As each analysis group has different steps of analysis but some of them are similar it makes sense to us to provide the analysis methods in a table format. The data is accessible via GFZ Data Services.

**3) What validation has been done between the spectral signature of a sample and its geochemistry?** None, in this paper we only describe the data that is accessible, no interpretation whatsoever is provided either here or in the article.

**3. Authors responses to David Turners - Reviewer Comments:**

We respond here to the comments that needed a more detailed answer other than addressing them in the publication itself.

**I) Datapaper + Supplements – general comments:**

**1) While it is clear that there are linkages between the datasets, there is not linkage between all the topics. For example, the REE-related spectra match well with the Cu-related spectra, but not the Apliki site data. It is this reviewer's opinion that two ESSD papers could be written, perhaps best to match the REE and Cu mineral-focused spectra together in a single paper, and the Apliki site data by itself. That way, the target information is more consistent within the described datasets. In the present format, the four suites of data are all different in the physical nature of the samples and the (geo)chemical and mineral characterization.**
We are able to comprehend the reviewers argumentation and realize the difference between the data sets presented here. As we are planning to publish more spectral data of different origin and sample characteristics, this article is mainly focussing on a comprehensive overview of the spectral data acquisition in the GFZ laboratory . The datasets themselve do not have to have the same geochemical or mineral characterization, as their physical state and geochemistry is separately described in the data reports themselves. The introduction has now been restructured to introduce the datasets more clearly and explain the aim of presenting different data collections.

**2) The manuscript starts with some fairly sweeping comments in the introduction that could be toned down in the context of this contribution.**
You're completely right! We acknowledge the enthusiastic tone relating to our contribution to the accredited USGS libraries and toned it down. The rest of the introduction is a review of the associated methods and to the best of our knowledge in a neutral tone. When re-reading the manuscript we tried to restructure section parts that detracted the reader.

**3) In order for the broader community to use any spectral library, documentation needs to be clear and there should be no ambiguity regarding data source or characterization methods.**
We have taken your comments regarding the document seriously and cleared up the document based on your indications regarding the state of purity of the spectra, the mineral structure and the different geochemical analysis types. This should help the reader to clearly catch on the scope of this work. The aim was to describe the hyperspectral data acquisition in the laboratory that is consistent for all sample types. The geochemical analysis differences have been explained in detail (in the main paper and the data reports) and are dependent on the target material.

**4) In this sense, I also recommend splitting the paper into two discrete contributions so as to reduce the potential confusion over what methods were used on which samples.**
It is our aim to show the hyperspectral data acquisition and explain in detail how the hyperspectral laboratory work is conducted. All datasets are united and comparable by the method of hyperspectral data acquisition in the laboratory. We have added substantial new information to the Apliki Technical Report that makes the information on samples and spectral libraries of Apliki more comparable to the other two spectral libraries. As mentioned above, the main aim of this manuscript is to describe the method that was used to create all three spectral libraries.

**5) Furthermore, the samples that are rocks and not minerals need to be identified as such. For example, the monazite and synchysite spectra are not what I would have expected if these were mineral specimens. Since one of the principle uses of these spectra are as inputs to understand unknown spectra in other datasets, it is important to state their true nature.**
We cleared this in the data report as well as in the ESSD paper. The REE-bearing minerals/ minerals within a rock matrix. All of these samples are supplied with XRF data but not all were measured with SEM. As these samples have been part of a number of diverse publications within the Ph.D. project of N.K. Bösche, we did not re-analysis the samples. As the results based on these data (Bösche, 2015) and successful publications we wanted to share them with a broader community. We have now excluded the Monazite sample and all corresponding information from the data, resulting in 13 REMin samples instead of 14.

**II) Datapaper Comments**
**Line 70. REE oxides are not minerals, though they are likely crystalline. Do you know the structure of these REE Oxides?**

You are correct, the REE-oxides are synthetic powders not in crystalline structure. The technical report now states this clearly.

**Line 104. How did you validate the mineral species?  (is related to the REE-bearing minerals)**
For the REE-bearing Minerals the notation from the supplier was assumed valid. The supplier (http://www.seltene-mineralien.de) offers analytical services with a modern REM-EDX technology and therefore, we assume the specimen he  analysed and the mineral species were validated before the sale.

**Line 108. How did you validate the mineral species?  (is related to the Copper-bearing minerals)**

Mineral denomination is based on microprobe analyses.

**Table 1. Why two suites for Copper bearing minerals? State why.**

The supplements were deleted, the copper bearing minerals are described in a revised table in the technical report.

**Line 345. 120 seconds isn't really short for a HHXRF**

The long duration of 120 seconds were chosen with the purpose to reduce the noise of the measurement.

**Line 121. "The sample preparation varied by sample type and depends on the material and the information of interest." This is a bit problematic. It is not that any one of the approaches isn't valid, but the presentation of the data as a collection should strive to have internally consistent methods.**

The hyperspectral data acquisition described in this document is the internally consistent method that is presented here. We are planning to publish more hyperspectral libraries in the future. All of these hyperspectral libraries will be compiled under the conditions as described here. The geochemical analysis is different not only based on the sample type but the level of detail required for the projects where our spectral libraries stem from. The geochemical analysis is presented only as a source for the users to check or interpret the hyperspectral data and not as a data publication itself. Unfortunately our funding does not allow for the compilation of spectral libraries with consistent geochemical validation as is possible for  the USGS Spectral Library Version 7 (Kokaly et al., 2017).

**Section 5.2 Why do you have two distinct descriptions for the JEOL unit at Potsdam?**

The University of Potsdam used two JEOL units, one for SEM and one for EMPA analyses. The EMPA analyses are described twice, one each for the sample type analyzed, as different measurement parameters were used.

**Line 272. Which standards? Same as above? If so, many of the Smithsonian materials are synthetic.**

SEM measurements are calibrated with pure copper and the EMPA measurements are calibrated with the standards mentioned in the text from the Smithsonian Institution and Astimex.

**III) Apliki technical report:**
**To confirm above points, the "aquatic" sample above is not an aquatic sample. Instead, it was analyzed by aqua regia digestion, as described in the BVM analytical notes. Why was this one sample analyzed with a different method? The manuscripts need to have this item addressed.**

The internal BVMs sample preparation-/ analysis type groups by the required analysis method. These "analysis types" are namely "aquatic", "rock" and "soil". The measurement/ analysis type is chosen by the service provider "BVM Institute" on which we did not have any influence.

**It is unfortunate that your overlimit Cu samples were not re-analyzed for total Cu content, since this is a key focus for the Cu mine related samples.**

Yes, the BVM analysis limits were chosen in order to resolve sample copper content with lower limits. We honestly did not expect to have three samples with Cu content > 10.000 ppm. Unfortunately, the analysis could not be repeated.

**Figure 2. Maybe you could split this out into two plots, rocks and soils, so that the reader can maybe make a small assessment of the data without downloading and plotting the spectra?**

We've given the plot a larger space in the document and added a subset of the spectral library to showcase some of the spectra. Additionally the spectra are presented and shown in more detail in the Ph.D. thesis of Koerting (2021, awaiting defence).

**IV) Copper-bearing technical report:**

**Sample C3 has a typo in its name, where a "%" is used instead of "5"**

Unfortunately, we cannot find the typo, maybe it was fixed during our reworking of the document from review 01

**There are some samples that are not monomineralic, and this is important to state in the documentation. For example, "Malachite 1" has a whack of other minerals in the SEM image, so right away we know that the "Malachite" spectrum here is actually at least Malachite+pyroxene+quartz. Same for Azurite 2 and Chalcopyrite 3.**
You are absolutely right, the sample names were used from the names of the collection where the samples were retrieved from. We added this information to the technical report. All identified mineral phases of each sample are presented in the file "copper_bearing_minerals_chemistry.pdf" in the published data. The samples in question are: A1_Azurite, A3_Azurite, L1_Linarite, P3_Plancheite, B1_Brochantite, M2_Malachite & C3_Chalcopyrite

**V) REE technical report:**
**You should include the anticipated crystal structure of these oxides. The work by White (1965?... off the top of my head) showed the importance of this variable for the resulting spectra. This is especially relevant in the context that Tb is listed as both 3+ and 4+.**

The synthetic REO powders were delivered by the supplier. The REE-bearing minerals and REO powders were published by Bösche (2015) and Herrmann (2019) and are described in detail in the reviewed works.

**Similar to the copper minerals, some of these samples are not monomineralic and instead are rocks, and therefore the spectra need to be identified as such. For example, the images you show in the EMPA data indicate that synchysite has a bunch of phases, as does ilmenite.**
You are absolutely right, the spectra of the REMin were supposed to be acquired only over the identifiable mineral surface in the hyperspectral imagery. The geochemical data indicate though, that some of the sampled area is not monomineralic. The geochemical data is supplied with the samples to show exactly that. We will note this in the geochemical part in the technical report.

**Do you have analyses of the monazite sample?**

Yes, we provided the XRF analyses of the sample in Version 1 of the technical report + data. After a revision, we, however, decided to exclude the Monazite from the data in the new version of the data and the technical report.